# High capacity of integrated crop-pasture systems to preserve old soil carbon evaluated in a 60-year-old experiment

Maximiliano González-Sosa[1,2], Carlos A. Sierra[2], J. Andrés Quincke[3], Walter E. Baethgen[4], Susan Trumbore[2], M. Virginia Pravia[5, 2]

1 Universidad de la República, Facultad de Agronomía, Departamento de Suelos y Aguas, Montevideo, Uruguay.
2 Max Planck Institute for Biogeochemistry, Jena, Germany
3 Instituto Nacional de Investigación Agropecuaria, INIA - La Estanzuela, Colonia, Uruguay
4 International Research Institute for Climate and Society, The Earth Institute, Columbia University, New York, USA
5 Instituto Nacional de Investigación Agropecuaria, INIA – Treinta y Tres, Treinta y Tres, Uruguay

*Correspondence to:* Maximiliano González Sosa (mgonzalez@fagro.edu.uy)

**Abstract.** Integrated crop-pasture rotational systems can store larger soil organic carbon (SOC) stocks in the topsoil (0-20 cm) than continuous grain cropping. The aim of this study was to identify if the main determinant for this difference may be the avoidance of old C losses in integrated systems, or the higher rate of new C incorporation associated with higher C input rates. We analyzed the temporal changes of 0-20 cm SOC stocks in two agricultural treatments of different intensity (continuous annual grain cropping and crop-pasture rotational system) in a 60-year experiment in Colonia, Uruguay. We incorporated this information into a process of building and parameterizing SOC compartmental dynamical models, including data from SOC physical fractionation (POM > 53 µm > MAOM), radiocarbon in bulk soil and $CO_2$ incubation efflux. This modeling process provided information about C outflow rates from pools of different stability, C stabilization dynamics, as well as the age distribution and transit times of C. The differences between the two agricultural systems were mainly determined by the dynamics of the slow-cycling pool (~ MAOM). The outflow rate from this compartment was between 3.68 and 5.19 times higher in continuous cropping than in the integrated system, varying according to the historical period of the experiment considered. The avoidance of old C losses in the integrated crop-pasture rotational system resulted in a mean age of the slow-cycling pool (~ MAOM) over 600 years, with only 8.8% of the C in this compartment incorporated during the experiment period (after 1963) and more than 85% older than 100 years old in this agricultural system. Moreover, half of the C inputs to both agricultural systems leave the soil in approximately one year due to high decomposition rates of the fast-cycling pool (~ POM). Our results show that the high capacity to preserve old C of integrated crop-pasture systems is the key for SOC preservation of this sustainable intensification strategy, while their high capacity to incorporate new C into the soil may play a second role. Maintaining high rates of C inputs and relatively high stocks of labile C appear to be a prerequisite for maintaining low outflow rates of the MAOM pool.

## 1 Introduction

Soil organic carbon (SOC) is currently at the center of international discussion as a relevant property of soils to address global issues such as food security and climate change (Lal, 2018, 2016). Firstly, it is the primary indicator of soil quality, because of its direct relationship with the physical, chemical, and biological properties that determine soil fertility and productivity (Reeves, 1997). Additionally, soils contain approximately two times more C than the atmosphere (Jobbágy and Jackson, 2000; Janowiak et al., 2017), and therefore, slight increases in their storage have the potential to reduce atmospheric $CO_2$ levels and contribute to the fight against climate change (Fargione et al., 2018).

SOC is a heterogeneous mixture of different components that decompose at different rates (Kögel-Knabner et al., 2008), and modeling SOC dynamics as a single pool overestimates the system's response on time scales of decades to centuries (Trumbore, 2009). In this context, the separation of SOC pools with different kinetics is fundamental for the accurate representation of their dynamics (Lavallee et al., 2020). The separation into particulate organic matter (POM) and mineral associated organic matter (MAOM) (Cambardella and Elliott, 1992) is one of the fractionation techniques that has proven to be highly effective. POM is composed of low-density materials, with little microbial processing and chemical characteristics close to the plant input material, while MAOM is a fraction protected from decomposition through association with the mineral phase, where individual molecules or small fragments of organic matter predominate with a greater contribution of microbial-derived compounds (Lavallee et al., 2020). Because of the different stabilization processes that characterize each of these fractions, on average MAOM tends to have lower decomposition rates (longer persistence) than POM (Poeplau et al., 2018; Trumbore and Zheng, 1996; Heckman et al., 2022). However, although they constitute a good proxy for characterizing compartments with different kinetics (Poeplau et al., 2018), they are not completely homogeneous compartments, and POM may include some proportion of slow cycling C, while MAOM may include some proportion of fast C.

There is a growing scientific consensus that the genesis of persistent SOC compartments (i.e. carbon that persists for decades to centuries or even longer once added to soil) occurs through the association of microbially synthesized products with the mineral phase (Cotrufo et al., 2015, 2013; Kallenbach et al., 2016; Kleber et al., 2011). Results from different studies suggest that higher levels of microbial anabolic activity under more diverse plant communities with perennial components are responsible for the formation of SOC at higher rates (Ma et al., 2018; Zhu et al., 2020), with a strong relationship between the production of microbial necromass and the formation of stable organic matter (Córdova et al., 2018; Zhu et al., 2020). These processes are particularly enhanced in soils such as Mollisols that, due to their fine textures, promote the stabilization of microbial derived C in association with the mineral phase (Cotrufo et al., 2013).

Low carbon inputs and intensive tillage have been identified as some of the main causes of soil deterioration and losses of soil organic carbon from agricultural systems (Rui et al., 2022). Increasing SOC stocks requires either increased carbon inputs (e.g. King and Blesh (2018)) without compensatory SOC losses, or decreased SOC losses relative to inputs. Various management practices such as reduced tillage, crop diversification, and application of amendments, have proven to be effective in increasing the poorly transformed particulate fractions of topsoil (0-30 cm) organic matter, but their effectiveness in generating more

persistent SOC in association with the mineral phase has been debated (Ogle et al., 2012; Rui et al., 2022). Nevertheless, the incorporation of perennial pastures into agricultural rotations also has proven to be a sustainable intensification strategy (Baethgen et al., 2021; Davis et al., 2012; Pravia et al., 2019). Plant covers such as perennial pastures, characterized by a high partitioning of C towards their root systems (and rhizodeposition), could constitute an important tool to increase soil C sequestration (Sokol and Bradford, 2019). Previous studies have found that increases in SOC stock associated with the inclusion of perennial pastures in rotations are linked to an increase in C inputs (King and Blesh, 2018), but the potential effects of changes in input quality are less clear (King et al., 2020; Macedo et al., 2022). In systems with perennial pastures, microbial anabolism processes would be maximized by the presence of living roots releasing organic products within the soil (Schmidt et al., 2011; Sokol and Bradford, 2019; Villarino et al., 2021). In this regard, a previous study shows that integrated crop-pasture systems are able to maintain high soil C stocks compared to grain cropping systems (Baethgen et al., 2021), but the processes that determine this dynamic continue to be poorly understood.

The study of C isotopes, and particularly $^{14}$C, is a useful tool to understand the dynamics of C exchange between terrestrial ecosystems and the atmosphere (Torn et al., 2009). Radiocarbon is a measure of the time elapsed since C was fixed from the atmosphere via photosynthesis, and therefore, is an extremely useful tool to estimate the cycling rates of different compartments of C in terrestrial ecosystems (Trumbore, 2009). In the particular case of SOC, which constantly receives new C inputs via photosynthesis and loses it through decomposition, the $^{14}$C isotopic signature of SOC reflects the $^{14}$C of the C input, the decomposition rate, and the radioactive decay rate of this isotope (Trumbore, 2000). The atmospheric radiocarbon peak produced by nuclear weapons testing up to the 1960s serves as an isotopic tracer to study SOC dynamics (Torn et al., 2009; Trumbore, 2009). The combined use of $^{14}$C measured in bulk soil and in incubations allows for the determination of the rate at which new C is incorporated and to estimate the possible C sources for heterotrophic respiration (Nowinski et al., 2010; Torn et al., 2013). This information is highly useful to identify whether C losses as oxidation derive from active compartments with signatures close to the input (i.e. that of the atmosphere at a certain point in time), or if, on the contrary, there is a significant contribution of C from compartments of old organic matter that are undergoing destabilization processes.

Although the effect of management on SOC stocks is a question that has received significant research effort, the effect of environmental triggers (i.e., climate, agronomic management) on the persistence of SOC is a fundamentally relevant aspect that began to be addressed only recently (Lehmann and Kleber, 2015; Schmidt et al., 2011; Sierra et al., 2018b). Based on information about C flows, stocks, and isotopic tracers such as radiocarbon, mathematical representations can be generated in the form of systems of differential equations that represent C dynamics in soil (Sierra et al., 2018a). Then, once the model correctly represents the measured data, emergent properties of the system such as the age and transit time can be derived (Sierra et al., 2018b).

In this work, we seek to improve the understanding of the biogeochemical processes that determine the success of integrated crop-pasture rotational systems in terms of their carbon storage and sequestration capacity compared to continuous annual grain systems. The general objective was to identify if integrated crop-pasture rotational systems can store larger SOC stocks than continuous cropping agriculture because of the avoidance of old C losses or due to a higher rate of new C incorporation

associated with higher input rates to the system. For this purpose, we analyzed the temporal changes of SOC in two agricultural treatments of different intensity (continuous cropping and crop-pasture rotational system) in a 60-year experiment in Colonia, Uruguay. We incorporated this information, data from SOC physical fractionation, radiocarbon measurements in SOC and $CO_2$ incubation efflux into a process of building and parameterizing SOC compartmental dynamical models. These models allow the analysis of C flow over time through compartments of different cycling rates and can be interpreted as processes of SOC (de)stabilization responsible for the observed differences between the analyzed systems. Finally, we used these models to derive the distribution of age and transit time of C in each agriculture system. To address the objective, the following possible mechanisms explaining the higher C storage in integrated crop-pasture rotational systems compared to intensive agriculture were hypothesized: 1) large input rates that promote SOC stabilization processes that support a high SOC stock (hypothetical MAOM accrual mechanism); 2) large input rates that promote the accumulation of large stocks of poorly stabilized particulate C (hypothetical POM accrual mechanism); 3) high persistence of very old SOC linked to low oxidation rates of passive SOC pools (hypothetical MAOM persistence mechanism); 4) a combination of the previous processes.

## 2 Materials and methods

### 2.1 Experimental site

In 1963, the National Institute of Agricultural Research of Uruguay (INIA) established a long-term agricultural experiment (LTE) at La Estanzuela experimental station (Colonia, Uruguay; 34°20'33"S, 57°43'25"W), located at the center of the Río de la Plata Grassland Ecoregion (Baeza et al., 2022). The site has a humid temperate climate, with an annual mean rainfall of 1126.65 ($\pm$ 269.94) mm and an accumulated reference evapotranspiration of 1192.54 ($\pm$ 74.30) mm over 53 years of meteorological records. The annual average temperature is 16.88 °C, with monthly average maximum temperatures of 29.04 °C and 14.83 °C, and monthly average minimum temperatures of 17.83 °C and 6.24 °C in January and July, respectively. The variations in temperature and evapotranspiration between winter and summer are significant, and there is a trend towards lower average accumulated rainfall volumes during the winter (Fig. A1a). Rainfall is highly variable among years, but it does not show any long-term trend (Fig. A1b). The climatic characteristics of the region allow the establishment of summer and winter crops, and typical agricultural systems consist of four crops in three years (Baethgen et al., 2021).

The geomorphology of the research site consists of rolling hills with an average slope of 3%. Soil is moderately acidic with medium to high natural fertility and presents a well-developed Bt horizon and no rockiness. It is classified as a Haplic Phaeozem (Vertic, Eutric) in the FAO soil classification system (IUSS Working Group, 2014), and as a fine, smectitic Vertic Argiudoll in the USDA Soil Taxonomy System (Soil Survey Staff, 2014). In 1985, a soil survey at the site reported SOC mass fraction of 20.8 g kg$^{-1}$, N mass fraction of 1.7 g kg$^{-1}$, 28.7% clay, and 63.7% silt for the 0-30 cm depth (Table A1).

## 2.2 Long-term experiment

The INIA La Estanzuela LTE is one of the oldest agricultural experiments in the world. It was set up with the original objectives of evaluating the impact of N and P fertilization in continuous cropping systems, as well as the effect of incorporating pastures in rotation with crops as a N source and technological alternative to prevent soil quality loss (Díaz and Morón, 2003). The LTE assesses the effect of seven treatments that represent a gradient of agricultural intensification on crop productivity, soil properties, and environmental impacts (Baethgen et al., 2021; Grahmann et al., 2020). This gradient ranges from continuous

cropping systems (with and without fertilizer application) to more conservationist agriculture in rotation with perennial pastures in different proportions of the rotation period.

The seven treatments are arranged in a randomized complete block design with three replications, and each of the 21 plots has an area of 0.5 ha. Currently, crop rotation sequences are not synchronized among the three replications of each treatment to have partial control of the year effect in the experimental design. Since 1963, the experiment has undergone a series of

adjustments aimed at more accurately representing the evolution of Uruguayan production systems. These modifications involved minor changes in the crops included in each of the rotations and, mainly, changes in soil preparation systems with a trend towards no-tillage. In the first 20 years of the experiment, soil preparation was carried out with conventional tillage (moldboard and disk plow) for the establishment of all crops and pastures in all treatments. Starting in 1983, the use of a chisel plow was gradually adopted, and pastures were sown in association with the last crop in the integrated crop-pasture treatments

to avoid tilling intervention.  From 2009 onwards, no-till farming was adopted in all treatments, eliminating the mechanical operations of soil preparation. Before the establishment of the LTE in 1963, the site had been cultivated for over 40 years, mainly with wheat (*Triticum aestivum* L.) – fallow systems. More detailed information about the LTE can be found in: Baethgen et al. (2021), Díaz and Morón (2003), Grahmann et al. (2020) and Quincke et al. (2019).

In order to maximize the differences in SOC stocks and dynamics, two contrasting treatments were selected: i) continuous

annual grain system (CC) and ii) a crop-pasture rotation system with a 50% agricultural phase and 50% pasture phase (R). Currently, the CC system includes a sequence of barley (*Hordeum vulgare* L.) or wheat (*Triticum aestivum* L.) in winter alternating with corn (*Zea mays* L.), sorghum (*Sorghum bicolor* L. Moench), or soybean (*Glycine max* (L.) Merr.) in summer. The R system includes three years of the crop sequence from CC, rotating with three years of a perennial pasture that consists of a mixture of legumes: white clover (*Trifolium repens* L.) and birdsfoot trefoil (*Lotus corniculatus* L.) and grasses: tall fescue

(*Festuca arundinacea* Schreb.). A detailed description of the crop sequences of these two production systems, as well as the modifications that occurred at different periods, can be seen in Fig. 1.

Winter crops (wheat, barley) were sown from May to June and generally harvested in December, while summer crops (sunflower, corn, sorghum, and soybean) were sown between November and December and harvested between April and May. The annual crops were harvested for grain production, while the plant residues were left on site. On the other hand, the pastures

were mowed to simulate grazing, and the forage was left in the experiment. In both systems, fertilization (N and P) of crops and pastures is carried out according to recommendations based on soil and plant analysis.

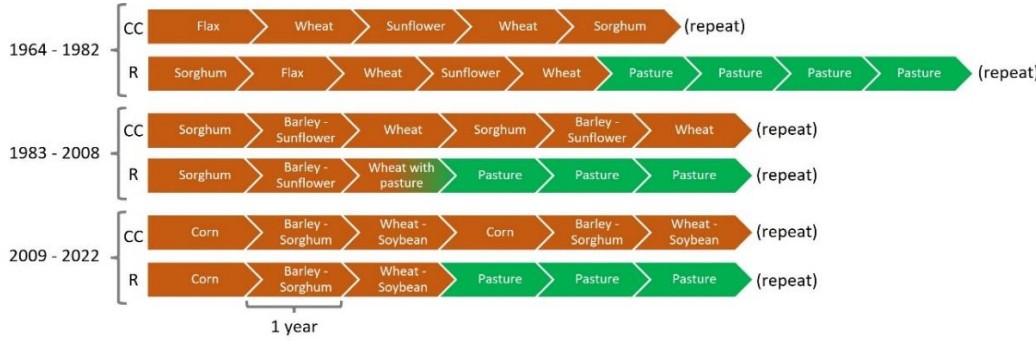

**Figure 1.** Crop sequences of the CC and R treatments in the different historical periods of La Estanzuela LTE.

## 2.3 Soil Sampling and Analysis

All 21 plots were sampled annually to a depth of 20 cm from 1964 to 1996 and to a depth of 15 cm from that year onward. As outlined in a previous study conducted on this experiment (Grahmann et al., 2020), changes in the sampling depth are not expected to affect the results of long-term trends since the soil was systematically homogenized by tillage to a depth of 20 cm before sampling until 2009. Taking this into account, we extrapolated the information on SOC stock measured from 1996 onwards up to a depth of 20 cm by multiplying by a coefficient equal to 20/15. Each composite sample consisted of 20 subsamples taken from the center of each plot. Samplings were performed during the fallow periods prior to the sowing of winter crops (April - May).

These samples were processed at the Laboratory of Water, Plants and Soils of INIA La Estanzuela Experimental Station (Colonia, Uruguay). The samples were oven-dried at 40 °C, ground and sieved to less than 2 mm before being analyzed to determine carbon content using $K_2Cr_2O_7$ and heat with the method described by Tinsley (1950) until 2011, and by dry combustion at 900 °C followed by infrared detection from 2012 on a LECO analyzer (Wright and Bailey, 2001). The values obtained with LECO were adjusted by a coefficient of 0.81 to be comparable with the previous time series. This coefficient derived from an internal laboratory validation process in which a regression model was built with more than 2000 samples. The bulk density measured in 2021 for the 0-20 cm layer of each plot of the analyzed treatments was used for the calculation of SOC stocks (R: $1.28 \pm 0.02$ Mg m$^{-3}$; CC: $1.38 \pm 0.03$ Mg m$^{-3}$).

Additionally, a stratified sampling was carried out at 0-10 cm and 10-20 cm in 2008 and 2021. These samples were analyzed to obtain the radiocarbon signature in the bulk soil and in $CO_2$ efflux from incubations (only 2021 samples) at the Accelerator Mass Spectrometry (AMS) Laboratory at the Max Planck Institute for Biogeochemistry (MPI-BGC, Jena, Germany). We incubated the soil in hermetic glass bottles at 25 °C and with a moisture content equal to 60% of the soil field capacity to promote heterotrophic respiration. A pre-incubation was carried out to avoid effects of the $CO_2$ flush from disturbance. Air was extracted from the headspace of each bottle once enough C had accumulated for graphitization and subsequent radiocarbon measurement (1.8 to 2.0 mg of C). To reach this threshold, the $CO_2$ concentration in the bottles was monitored with an infrared

gas analyzer (Li-6262). Air extraction from the incubation bottles was carried out on a vacuum extraction line that allows the cryogenic purification of $CO_2$ from other gases ($N_2$, $O_2$, $H_2O$) (Trumbore et al., 2016a). Low vapor pressure gases (i.e., water)

are trapped by passing the air flow through a trap submerged in a dry ice and alcohol bath (-78 °C). Then, the $CO_2$ is captured by making the air flow through a trap submerged in liquid nitrogen (-196 °C), while other gases ($O_2$, $N_2$) do not freeze at low pressures and are pumped away from the vacuum line (Trumbore et al., 2016a). Then, the purified $CO_2$ was reduced to graphite in the presence of $H_2$ as a reducing agent and iron powder (Fe) as a catalyst (Trumbore et al., 2016a) at high temperature (550 °C). The mixture of graphite and iron was analyzed for its radiocarbon signature at the AMS facility of MPI-BGC (Steinhof

et al., 2017).

Finally, the soil from the 2021 stratified sampling was physically fractionated by size, according to Cambardella and Elliot (1992), to represent C pools with different cycling rates, separating the particulate organic matter (POM, larger than 53 μm) from the mineral-associated organic matter (MAOM, less than 53 μm). POM and bulk soil samples were analyzed for C content by dry combustion (Elementar Vario Max) at MPI-BGC. The POM fraction was calculated considering its C concentration

and recovered weight and MAOM fraction was calculated as the difference between total SOC stock and POM stock. POM and MAOM C stocks were corrected by a coefficient of 0.81 to be comparable with the SOC time series.

## 2.4 Monitoring of C inputs

The plots were sampled annually to obtain crop grain productivity as well as the annual dry matter aboveground productivity

in the case of pastures, except for flax, which was estimated based on productivities reported for the site of the LTE between 1969 and 1971 (MGAP-CIAAB, 1971). Based on this information, aboveground dry matter production of crops was estimated using harvest indices extracted from Unkovich et al. (2010), Grant et al. (1999) and Mercau et al. (2007). Belowground C production was estimated considering aboveground production, shoot-to-root ratios (Bolinder et al., 2007) and net rhizodeposition to root biomass ratios (Pausch and Kuzyakov, 2018). The proportion of belowground C inputs corresponding

to the 0-20 cm layer was calculated by considering a coefficient of 0.72, obtained from the ratio between C POM (0-20 cm) and C POM (0-80 cm) for the site (data not shown) and assuming that the vertical distribution of roots correlates positively with the vertical distribution of C POM. In the case of pastures, a root turnover of 0.52 $y^{-1}$ (Gill and Jackson, 2000) was considered, except for the last year of the pasture phase for which a root turnover equal to 1 $y^{-1}$ was considered.

For all crops and pastures, a C concentration of 0.45 g g (dry weight)$^{-1}$ for all plant parts was considered (Bolinder et al., 2007).

Based on this information, soil C inputs were calculated as the sum of aboveground (subtracting grain production) and belowground C production for each period of the rotations. Accumulated inputs for each period of the crop rotations were then annualized to obtain the mean C input for each rotation period. Subsequently, an overall average for the entire historical series was obtained as the C inputs mean of all the periods for each system. All coefficients used in estimating soil C inputs are shown in Table 1.


**Table 1.** Coefficients used in calculations of soil C inputs based on crop and pasture yield data.

| Crop | Species | Harvest index | Shoot/root ratio[4] | Net rhizodeposition/root biomass[5] |
|------|---------|--------------|--------------------|------------------------------------|
| Wheat | *Triticum aestivum* | 0.37[1] | 7.4 | 0.54 |
| Barley | *Hordeum vulgare* | 0.38[1] | 7.4 | 0.54 |
| Sorghum | *Sorghum bicolor* | 0.46[1] | 5.6 | 0.54 |
| Corn | *Zea mays* | 0.49[1] | 5.6 | 0.54 |
| Sunflower | *Helianthus annuus* | 0.4[1] | 5.6 | 0.54 |
| Flax | *Linum usitatissimum* | 0.36[2] | 7.4 | 0.54 |
| Soybean | *Glicine max* | 0.5[3] | 5.2 | 0.54 |
| Pastures | | | 1.6 | 0.5 |

Note: [1] Unkovich et al., 2010, [2] Grant et al., 1999, [3] Mercau et al., 2007, [4] Bolinder et al., 2007, [5] Pausch and Kuzyakov, 2018.

## 2.5 Mathematical modelling

To analyze the systems as a whole and interpret the radiocarbon results concomitantly with all the available information, we used autonomous linear compartmental dynamical models with two pools of different decomposition rates and a transfer
scheme between them (stabilization) (Fig. 2).

In the models, we assumed a horizon that represents the 0-20 cm layer of the experimental plots. C enters the system through a fast cycling pool (representing POM C) and is then transferred to a slow cycling pool (representing MAOM C) at a rate determined by the cycling velocity of the labile pool and a C transfer coefficient between the two compartments.

The model is mathematically defined by a set of two differential equations, each of which describes the temporal changes of
C stocks in each pool:

(1)

$$\frac{dC_{fast}}{dt} = I - k_1 C_{fast}$$

$$\frac{dC_{slow}}{dt} = \alpha k_1 C_{fast} - k_2 C_{slow}$$

where I represents the amount of C inputs into the system; $C_{POM}$ is used to estimate the mass of C stored in the fast-cycling
pool ($C_{fast}$); $C_{MAOM}$ is used to estimate the mass of C in the slow-cycling pool ($C_{slow}$); $k_1$ and $k_2$ are the output rate constants from the fast and slow pool, respectively; α is the C transfer coefficient, representing the proportion of carbon leaving the fast pool that is transferred to the slow pool (C stabilization). This structure of a two-pool compartmental model with a connection in series has been used in works such as Spohn et al. (2023) and Stoner et al. (2021).

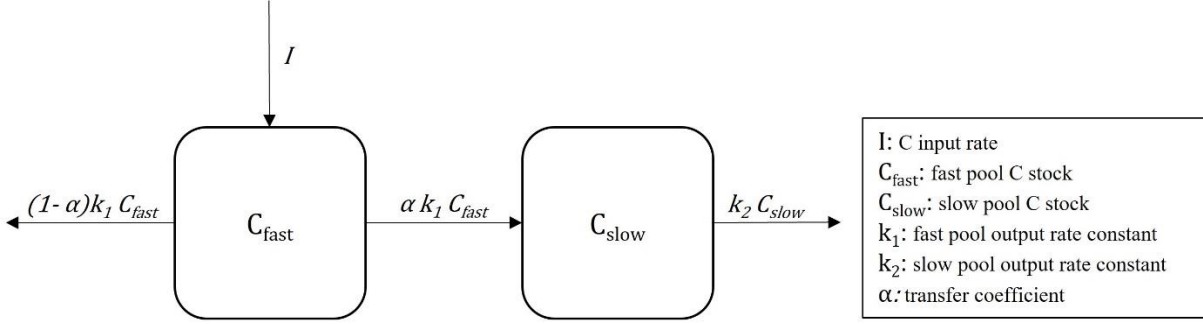


**Figure 2.** Graphical representation of the C compartmental model (Eq. 1) adjusted for CC and R agricultural systems.

The use of radiocarbon information as a tracer to improve the parameterization of this model is possible by the setup of a radiocarbon version of the model in Equation (1) that includes the rate of $^{14}$C radioactive decay and the fraction of radiocarbon

in the atmosphere (Eq. 2).

(2)

$$\frac{dF_{fast}C_{fast}}{dt} = F_a I - k_1 F_{fast} C_{fast} - \lambda F_{fast} C_{fast}$$

$$\frac{dF_{slow}C_{slow}}{dt} = \alpha k_1 F_{fast} C_{fast} - k_2 F_{slow} C_{slow} - \lambda F_{slow} C_{slow}$$

where $\lambda$ is the radiocarbon decay constant, $F_a$ is the atom fraction of radiocarbon in atmospheric $CO_2$, $F_{fast}$ and $F_{slow}$ are the

fractions of radiocarbon in the fast and slow pool, respectively. Radiocarbon is expressed as absolute fraction modern (Eq. 3), defined as the absolute ratio of the sample to a standard (OX-I, oxalic acid made from a sugar beet crop of 1955), corrected for radioactive decay to the year of measurement. (Trumbore et al., 2016b). By convention, the OX-I standard is normalized to a $\delta^{13}$C value of -19‰, and the measured sample to a value of -25‰ to correct for mass-dependent isotopic fractionation effects (Trumbore et al., 2016b).


(3)

$$F = \frac{\left.\frac{^{14}C}{^{12}C}\right|_{sample, -25}}{0.95 \left.\frac{^{14}C}{^{12}C}\right|_{OX,-19} e^{(y-1950)/8267}}$$

Fitting this model to the information from each of the agricultural systems (CC and R) allows testing the stated hypotheses. To achieve this, we will focus on evaluating the effect of the agricultural system on the C output rate constants of both pools

($k_1$ and $k_2$) and the stabilization coefficient $\alpha$. The time series of atmospheric radiocarbon data were obtained from Reimer et al. (2013) and Hua et al. (2022) for the pre and post bomb period (southern hemisphere), respectively.

The time series of annual SOC monitoring (1964-2022), the proportion of C by fraction (POM - MAOM) in 2021, the radiocarbon signature measurements in the bulk soil in 2008 and 2021, the radiocarbon signature in the $CO_2$ derived from incubations in 2021, and C input data derived from the productivity monitoring of the plots (section 2.4) were used to parameterize the model (Eq. 1 and Eq. 2) for each agricultural system (CC and R). The fitting was carried out to represent the temporal changes of SOC stock in the POM and MAOM fraction as the fast cycling (pool 1) and slow cycling (pool 2) model compartments, respectively.

Initially, an optimization procedure (Levenberg-Marquardt optimization algorithm) was applied to find the best set of parameters that minimizes the differences between predictions and observations. Then, this set was used as the starting point in a Bayesian Markov Chain Monte Carlo (MCMC) procedure to estimate the probability density distribution of the parameters (Soetaert and Petzoldt, 2010) and quantify the uncertainty associated with the generated models. For the MCMC procedure, a uniform distribution with a wide range of variation was used as a prior for each parameter, such that the outcome would be strongly dependent on the data. Similar modeling approaches have been used in studies such as Spohn et al. (2023), Crow et al. (2020), Crow et al. (2018), Sierra et al. (2013), Sierra et al. (2012b). The R packages SoilR (Sierra et al., 2014, 2012a) and FME (Soetaert and Petzoldt, 2010) were used to implement the models and parameterization procedures.

Based on the mathematical framework developed by Metzler and Sierra (2018), we conceptualized the deterministic models proposed in Equation (1) and Equation (2) as their stochastic versions and calculated the probability distribution of the transit time and the age of C in each of the systems and their pools.

The age of the system is defined as the age of the C atoms in the soil since the time they entered the system until the moment of observation and is a good indicator of the persistence of C in the soil, while the transit time refers to the time that elapses since each atom enters the system until it is released from it (Sierra et al., 2018b). These diagnostic characteristics allow for a global understanding of the system dynamics, emerging as integrating properties of all the processes represented in the model (Sierra et al., 2017), and can be used to identify the effect of environmental or management changes on the stability of C in the system as a whole.

The uncertainty of the fitted model for each agricultural system was propagated in a process whereby sets of parameters were iteratively drawn from the distributions obtained through the Bayesian fitting procedure (MCMC). These sets were used to run the models and calculate the ages and transit times in each iteration. By recording the values of these variables in each iteration of the previous process, their distributions were constructed, capturing the variability transferred from the parameter populations.

### 2.6 Statistical analysis

The effect of the agricultural system on the evolution of surface C stocks from 1964 onwards was analyzed using a mixed linear model (LMM) ($p < 0.05$) with the agricultural system and time as fixed effect factors and the experiment plot as a random effect factor to represent the association of measurements made on the same plot in different years. This analysis was

conducted using the lme4 package (Bates et al., 2015). The effect of the agricultural system on the stock of C in organic matter fractions (POM, MAOM), radiocarbon signature in the $CO_2$ efflux in 2021 and radiocarbon signatures in the bulk soil in 2008 and 2021, were conducted with paired t-tests ($p < 0.05$). We tested the significance of the differences between the compartmental model parameters and between the populations of ages and transit times through the comparison between agricultural systems of the 95% credible intervals for each of these variables (Makowski et al., 2019). All the analyses were performed in the R software (R Core Team, 2023).

## 3 Results

### 3.1 Measured C dynamics

We found a statistically significant effect of the agricultural system and time factors on SOC down to 20 cm (Wald chi-square test, p value < 0.001). A sustained decline in SOC was observed in the CC system, decreasing from $53.75 \pm 1.13$ Mg ha$^{-1}$ in 1964 to $41.85 \pm 2.49$ Mg ha$^{-1}$ in 2021 (Fig. 4b). In contrast, no trends of change were evident in the R system, where SOC has fluctuated around 56 Mg ha$^{-1}$ throughout the history of the experiment (Fig 4a). POM and MAOM C stocks showed significant differences between agricultural systems at all evaluated depths (Table 2).

Bulk soil radiocarbon was significantly different between systems in the 10–20 cm layer and the 0–20 cm layer for both measurement years (2008 and 2021), showing a trend of difference in the 0–10 cm layer in the year 2008 (p value = 0.077) and neither difference nor trend in this layer in 2021 (Table 2). It was observed that the soil at the evaluated depth is more depleted in radiocarbon in the CC system compared to the R system in both 2008 (CC $\Delta^{14}C_{0-20cm}$: -99.37 $\pm$ 17.24 ‰; R $\Delta^{14}C_{0-20cm}$: -40.0 $\pm$ 6.47 ‰) and 2021 (CC $\Delta^{14}C_{0-20cm}$: -72.11 $\pm$ 3.32 ‰; R $\Delta^{14}C_{0-20cm}$: -46.69 $\pm$ 6.65 ‰).

$CO_2$ radiocarbon measured in soil incubation experiments from 2021 samples showed significant differences between systems at all depths (Table 2), with values of $6.87 \pm 3.09$ ‰ in the CC system and $27.1 \pm 4.82$ in the R system (0-20 cm). In both cases, these were much more modern (closer to the atmospheric signature of the year of measurement) than the bulk soil. Calculated C input rates were $2.87 \pm 1.1$ Mg ha$^{-1}$ y$^{-1}$ and $4.94 \pm 1.9$ Mg ha$^{-1}$ y$^{-1}$ for the CC system and R system, respectively (Fig. 5, Table A3).

**Table 2.** Carbon stocks in fractions (POM, MAOM), radiocarbon in bulk soil and incubation efflux, and oxidation rates in incubations for each depth and agricultural system.

| System | Depth (cm) | Bulk soil $\Delta^{14}C$ (‰) | | Incubation efflux $\Delta^{14}C$ (‰) | Oxidation rate - microcosm (mg C h$^{-1}$ microcosm$^{-1}$) | Oxidation rate (mg g$^{-1}$ C h$^{-1}$) | POM stock (Mg ha$^{-1}$) | MAOM stock (Mg ha$^{-1}$) |
|---|---|---|---|---|---|---|---|---|
| | | 2008 | 2021 | 2021 | 2021 | 2021 | 2021 | 2021 |
| R | 0 - 10 | -34.17 [a] (3.57) | -47.27 [a] (9.54) | 9.13 [a] (2.98) | 0.0553 [a] (0.0029) | 0.080[a] (0.0052) | 5.55 [a] (0.48) | 22.40 [a] (0.23) |
| R | 10 - 20 | -46.9 [a] (9.9) | -45.97 [a] (2.97) | 48.3 [a] (5.56) | 0.0463 [a] (0.0033) | 0.089[a] (0.0034) | 1.16 [a] (0.12) | 20.90 [a] (0.79) |
| R | 0 - 20 | -40.0 [a] (6.47) | -46.69 [a] (6.65) | 27.1 [a] (4.82) | 0.051 [a] (0.0006) | 0.084[a] (0.00098) | 6.71 [a] (0.58) | 43.3 [a] (1.01) |
| CC | 0 - 10 | -91.5 [a] (17.60) | -60.4 [a] (3.71) | -6.5 [b] (3.15) | 0.0407 [b] (0.041) | 0.083[a] (0.0012) | 3.76 [b] (0.32) | 17.8 [b] (0.94) |
| CC | 10 - 20 | -107.7 [b] (16.86) | -86.47 [b] (2.85) | 25.63 [b] (3.77) | 0.0287 [b] (0.029) | 0.074[b] (0.0017) | 0.64 [b] (0.12) | 16.9 [b] (0.91) |
| CC | 0 - 20 | -99.37 [b] (17.24) | -72.11 [b] (3.32) | 6.87 [b] (3.09) | 0.035 [b] (0.0021) | 0.079[b] (0.00037) | 4.40 [b] (0.29) | 34.7 [b] (1.84) |

Note: Different letters within the same variable and depth indicate significant differences by paired t-test (p < 0.05) between systems. The value in parentheses indicates the standard error of the mean (n = 3). 'Oxidation rate – microcosm' corresponds to the total incubated material (25 g of soil); 'Oxidation rate' is standardized per unit mass of incubated C.

## 3.2 Model results

The results of the Bayesian calibration procedure (MCMC) for the model presented in Eq. 1 and Eq. 2 for each agricultural system, considering a surficial layer of 0-20 cm, are presented in Figure 3 and Table A2. To achieve a good model fit for the
335 CC system, it was necessary to build a two-stage model because a higher rate of SOC loss was observed from 1964 to 1990 (Period 1) compared to 1991 to 2021 (Period 2). The parameters $k_2$ and $\alpha$ were allowed to be adjusted independently for each period. It was assumed that the parameter $k_1$ did not vary between the two periods. The R system was assumed to be in steady state with respect to SOC stocks, as no trends of change were observed in this variable over time.

In the parameterized model, we observed that in both agricultural systems, the output rate constant of the fast-cycling pool ($k_1$) was two orders of magnitude higher than that of the slow-cycling pool ($k_2$). No statistically significant differences were found in $k_1$ between agricultural systems (Fig. 3). However, a clear difference between agricultural systems was observed in the output rate constant of the slow-cycling pool ($k_2$), which was approximately 5.19 times higher in Period 1 of the CC system compared to the R system, and 3.68 times higher in Period 2 of the CC system compared to the R system. No significant differences were observed in the $k_2$ parameter between both periods of the CC system, although the mean and median values reached lower values in Period 2. There was also no significant difference in the transfer coefficient $\alpha$ between agricultural systems or between periods of the CC system. The mean values for $\alpha$ were very low in all cases. This means that the transfer from $C_{fast}$ to $C_{slow}$ pool was small relative to the total $C_{fast}$ outflow (i.e., transfers to $C_{slow}$ plus decomposition). The mean values for $\alpha$ were $1.58 \pm 0.00027\%$, $1.69 \pm 1.77\%$, and $2.46 \pm 2.15\%$ of the C outflow rate from the fast pool in the R system, Period 1 of the CC system, and Period 2 of the CC system, respectively, which implies that most of the $C_{fast}$ pool (~ POM) was being respired rather than flowing to the $C_{slow}$ pool (~ MAOM).

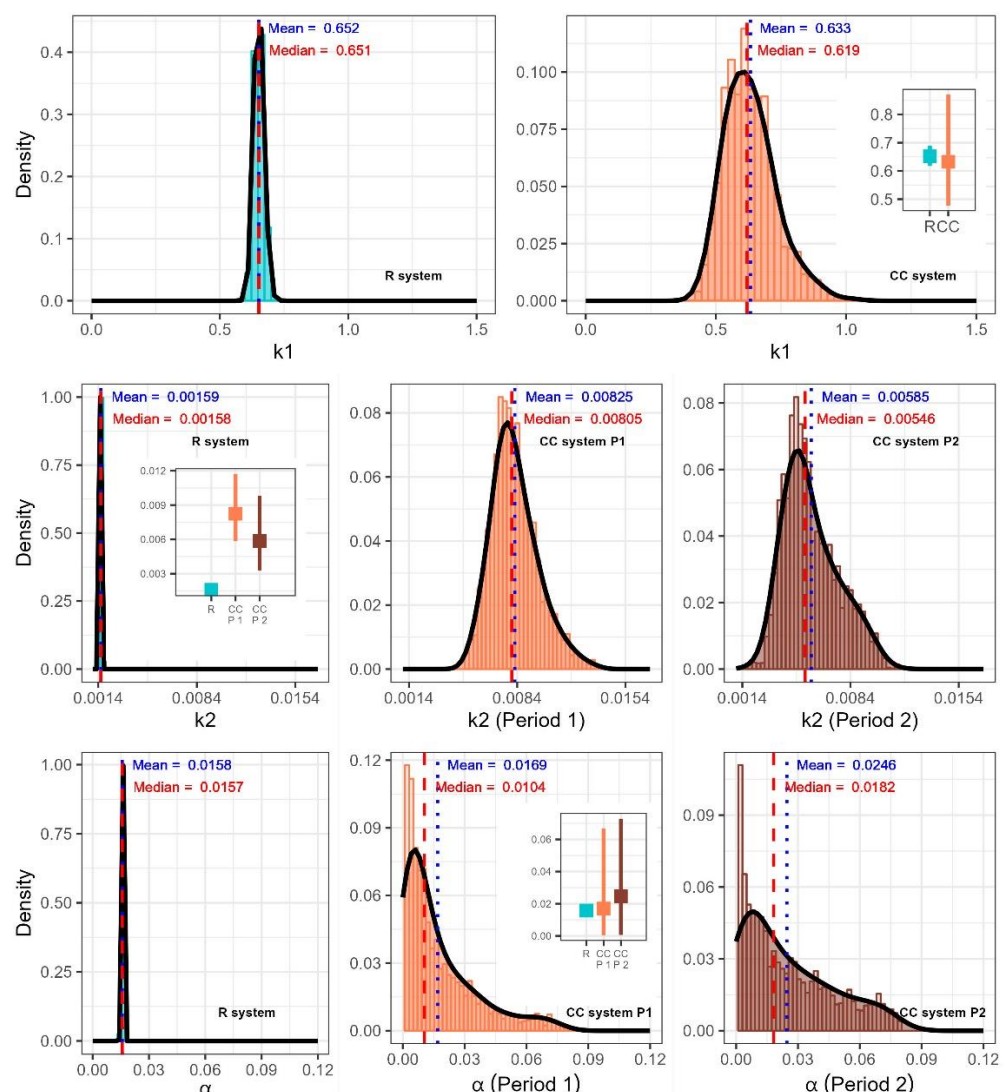

**Figure 3.** Compartmental model parameter populations for each agricultural system obtained through 25000 iterations for CC system and 10000 iterations for R system of the Bayesian fitting procedure (MCMC) (0-20 cm). The inset graphs correspond to the comparison of the 95% credible intervals for each parameter between agricultural systems (the value of the parameter is represented in y axis).

Figure 4 display the fit to the measured data of the parameterized model for each agricultural system. In all cases, a relatively good agreement is observed between the observations and the estimates. In the case of the R system, the assumption of steady state for C stocks appears to be reasonable, as no significant trends of change over time in the measured values are observed

(Fig. 4a). The model is also able to represent in an excellent way the measured radiocarbon data in the soil and in the system's outflow (Fig. 4c).

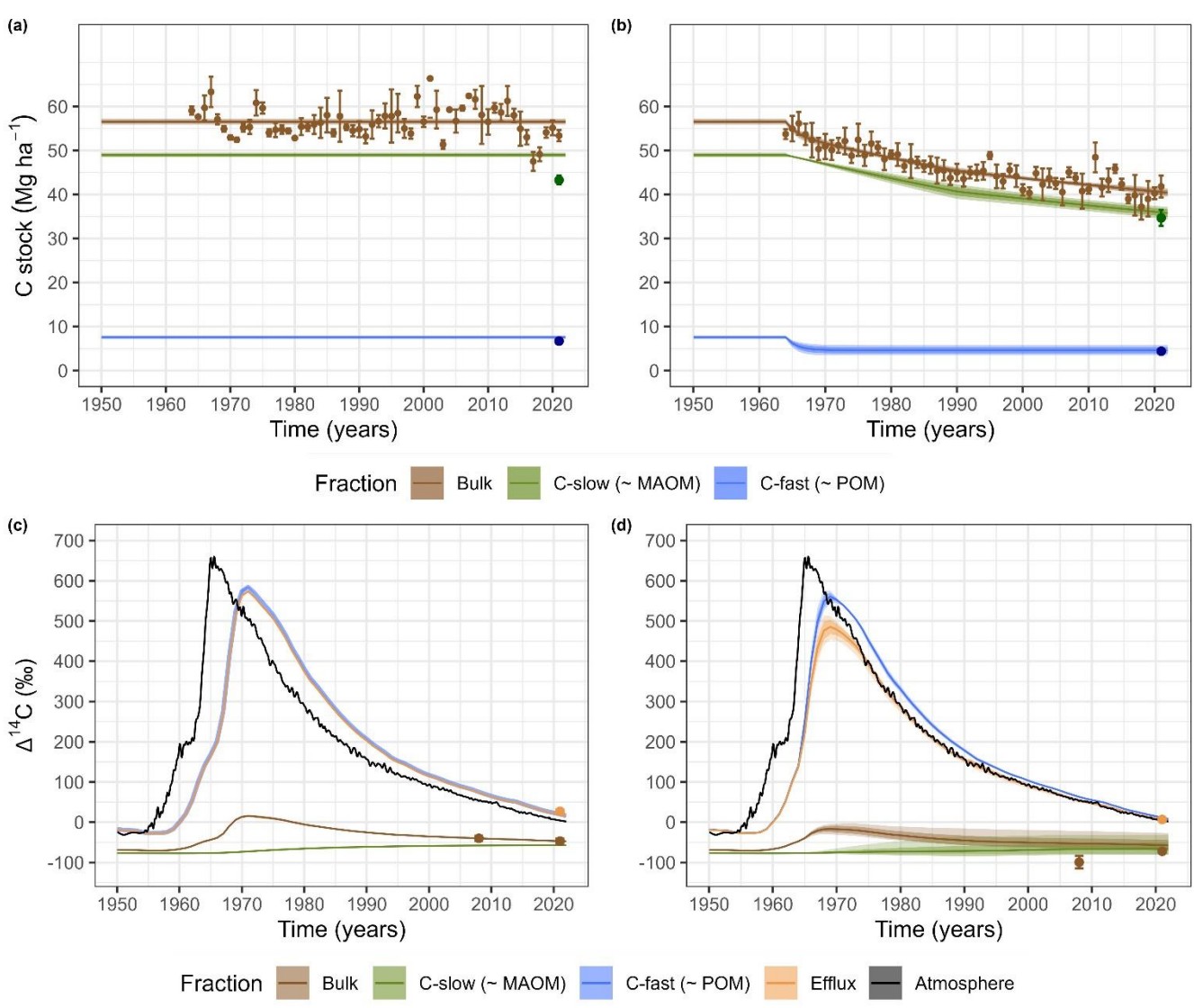

**Figure 4.** Model predictions of the temporal changes of C stocks (0-20 cm) in the bulk soil and by fraction for the R system (a) and CC system (b) and model predictions of the temporal changes of the radiocarbon abundance ($\Delta^{14}C$) since the bomb spike to the present for the R system (c) and CC system (d). Uncertainty ranges of model estimates obtained through sampling of the posterior parameter distribution (dark area: standard deviation; light area: 95% credible interval).

The main carbon flux rates for each agricultural system are presented in Fig. 5 and Table A3. It was assumed that the R system is in steady state with respect to SOC stocks, meaning that the total annual carbon outputs are equal to the inputs. For the CC system, the trajectory of carbon losses is determined by a higher rate of outputs than inputs. Additionally, the lower outflow rate constant of the slow-cycling pool in the R system resulted in this compartment having a lower relative importance in the total outputs (1.58%) compared to the CC system (6.66%). The stabilization flux (amount of carbon transferred from the fast pool to the slow pool per year) was 50 % higher in the R system, which was primarily determined by the difference in input rates between the two systems, as the stabilization efficiency (Stabilization flux / inputs) was roughly the same in both agricultural systems.

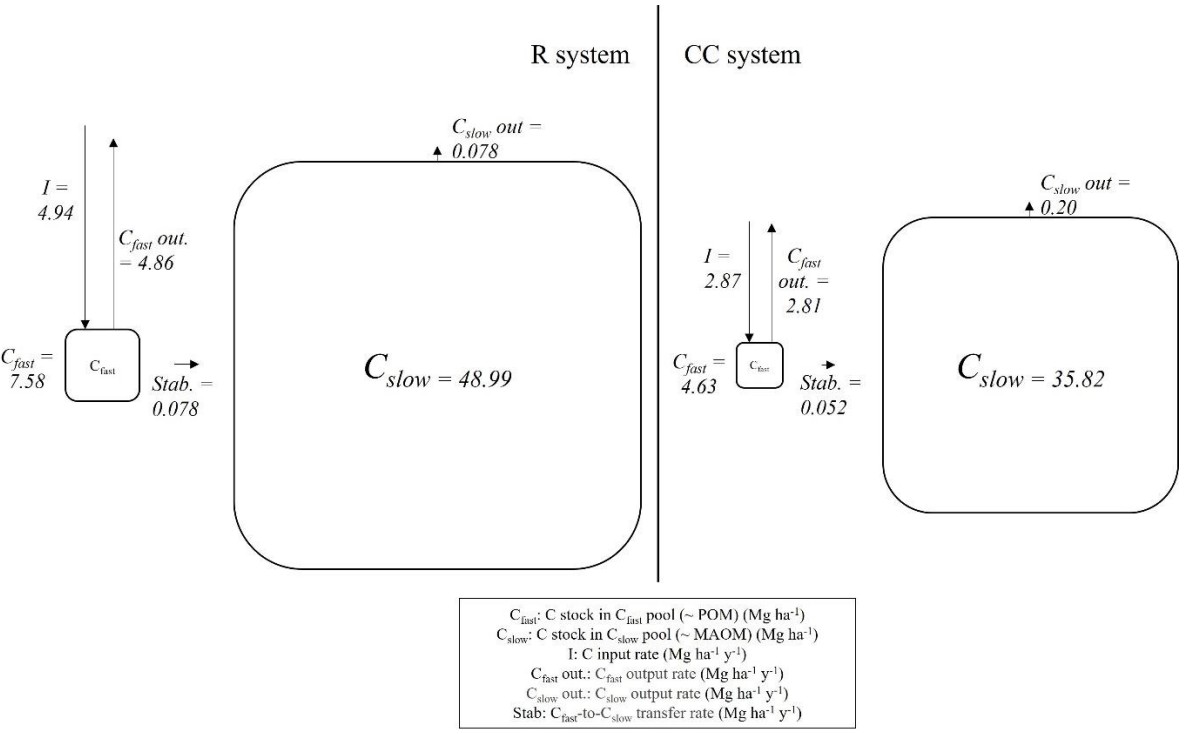

**Figure 5**. $C_{fast}$ and $C_{slow}$ stocks and main C flux rates derived from the compartmental model fitted to each agricultural system (steady state condition for R system and for 2021 in CC system)

The mean system age, defined as mean age of the atoms in the soil since the time they entered the system until the moment of observation (Sierra et al., 2018b) and calculated under steady-state conditions, was found to differ between agricultural treatments, with a value of $547 \pm 9.5$ years for the R system and $115 \pm 66$ years for the CC system (Fig. 6a). These ages are mainly determined by an older slow pool (~ MAOM) in the R system (mean age of $632 \pm 9.4$ years) compared to only $188 \pm$

56 years for CC system (Fig. 6c). Both the mean age of the system and the slow-cycling pool are significantly different between agricultural treatments, as no overlap of the credible intervals obtained through the propagation of the parameter distributions from the models is observed. The transit time distributions, defined as the distribution of the time that elapses since each C atom enters the system until it is released from it (Sierra et al., 2018b), obtained for the two agricultural systems (Fig. 6d) show that the respired carbon is predominantly of recent incorporation, with a mean transit time of $11.5 \pm 0.2$ and $6.2 \pm 4.4$ years in the R and CC systems, respectively. This relatively fast transit time is explained by a labile compartment ($C_{fast} \sim$ POM) with high outflow rate constant and young mean ages ($1.5 \pm 0.04$ years in the R system and $1.6 \pm 0.24$ years in the CC system) (Fig. 6b). Both the mean age of the fast-cycling pool and the mean transit time showed no significant differences between agricultural treatments. It is important to highlight that the age structure shown in this figure does not reflect the current condition of the CC system (which is not in equilibrium yet) but rather the age distribution it will reach once it achieves steady state if its current dynamics are projected forward.

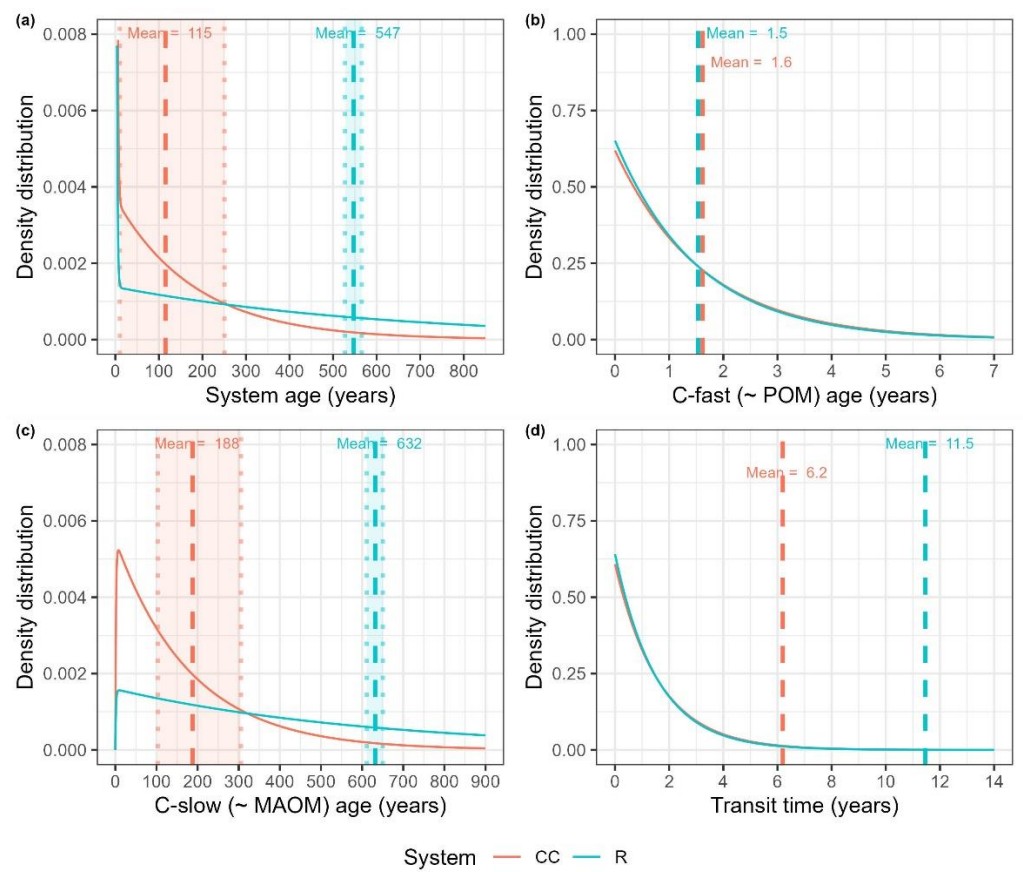

**Figure 6.** Distribution of C age in the system (a), in the $C_{fast}$ pool (b), in the $C_{slow}$ pool (c), and distribution of carbon transit time in the system (d) for each agricultural treatment. Dashed vertical lines indicate the mean of the distribution, and dotted vertical lines indicate the limits of the 95% credible interval of the mean (only intervals with significant differences are shown). For the CC system, the distributions are those that will occur once the system reaches a steady state if its current dynamics are projected forward.

## 4 Discussion

The integrative analysis conducted in this study, based on the parameterization of a compartmental model using monitoring information of C stocks, productivity, and radiocarbon measurements in bulk soil and incubations, derived from a 60-year agricultural experiment, allowed us to address the central research question concerning the processes responsible for higher C stocks in integrated crop-pasture rotational systems compared to continuous cropping.

The model exhibited a good fit to the measured data in both agricultural systems, as shown in Figure 4, except for the bulk soil radiocarbon in the CC system. Regarding this variable, while the model accurately represented the data for 2021, it overestimated the value for 2008. This discrepancy could be attributed to the fact that this representation does not explicitly characterize the soil erosion process, which might have displayed transient behaviors related to different soil preparation methods over the different periods of the experiment. Regarding this, it is important to highlight that tillage interventions were progressively reduced throughout the experiment starting from 1983, and soil management completely changed to no-till in 2009 in both systems. Therefore, erosive processes should have been more relevant in Period 1 (1964-1990) but significantly less in Period 2 (1991-2021). A previous study on the same experiment (Baethgen et al., 2021) suggests that the vast majority of SOC losses are due to heterotrophic respiration in both treatments (around 87% for the R system and 80% for the CC system). However, future studies should consider this process, explicitly separating SOC losses and evaluating their impact on radiocarbon dynamics and its use in model parameterization. Another possible explanation for the discrepancy between the CC system modeling and the radiocarbon data measured in 2008 is a potential bias because the fractions measured in the laboratory (POM – MAOM) may not exactly correspond to the kinetic pools ($C_{fast}$ – $C_{slow}$) defined in the model. This point could be tested in future studies with more organic matter fractionation data distributed over time and radiocarbon measurements of the POM and MAOM fractions. Another point that must be considered in interpreting the results of this work is that the available information only allowed us to consider the 0-20 cm layer as a homogeneous entity, both in terms of C inputs and SOC dynamics. This aspect could become relevant from 2009 onwards, when no-till system was adopted. Obtaining information with much more intensive sampling stratification schemes than those employed in this study could enhance the historical data of this experiment, enabling it to address new hypotheses in this direction.

As demonstrated in other similar modeling approaches (Kätterer and Andrén, 1999; Sierra et al., 2013; Spohn et al., 2023; Stoner et al., 2021), a two-pool structure proved to be adequate for representing these agricultural soil systems, characterized by a dominant slow-cycling compartment ($C_{slow}$ ~ MAOM) with an output rate constant ($k_2$) two orders of magnitude lower than that of the fast-cycling pool ($C_{fast}$ ~ POM) and a C transfer coefficient (α) ranging between 1.58 to 2.46% of the C outflow from the fast pool (Fig. 3, Table A2). This mathematical representation of C dynamics accounts for the great difference between the radiocarbon signatures of bulk soil and $CO_2$ in both agricultural systems. The former is highly negative and predominantly influenced by the signature of a predominant and persistent $C_{slow}$ pool (~ MAOM) that is relatively decoupled from the atmosphere, while the latter is very close to the atmospheric signature and is predominantly explained by the decomposition of the fast pool (~ POM) at high rates.

Integrated crop-pasture systems are capable of achieving higher SOC stocks in the topsoil compared to continuous annual grain system, as has been previously suggested (Baethgen et al., 2021; Pravia et al., 2019). The modeled SOC at steady state for the R system was $56.56 \pm 0.38$ Mg ha$^{-1}$ (Fig. 4a), whereas it was $40.45 \pm 0.58$ Mg ha$^{-1}$ (Fig. 4b) in 2021 for the CC system. Subsequent sections will discuss how different dominant processes hypothesized could account for the observed differences in SOC stock and dynamics.

POM Accrual Mechanism

Concerning the fast-cycling C, a modeled difference between agricultural systems in the C stock of the fast pool (~ POM fraction) for 2021 was found (2.95 Mg ha$^{-1}$), accounting for only 18.3% of the total SOC difference between the treatments (16.11 Mg ha$^{-1}$). Therefore, the difference in SOC dynamics between the analyzed systems is not predominantly explained by the dynamics of this pool. Furthermore, no statistically significant difference was observed between the two agricultural systems in the outflow rate constant of this compartment ($k_1$) (Fig. 3), which was $0.652 \pm 0.018$ y$^{-1}$ and $0.633 \pm 0.099$ y$^{-1}$ for the R and CC systems, respectively. Hence, the modeled $C_{fast}$ stock difference (R: $7.58 \pm 0.22$ Mg ha$^{-1}$ and CC: $4.63 \pm 0.70$ Mg ha$^{-1}$) is explained by the different mass of material entering to each of the systems through this pool (C input rate), which is 72% higher in the R system compared to the CC system (R: $4.94 \pm 1.9$ Mg ha$^{-1}$ y$^{-1}$; CC: $2.87 \pm 1.1$ Mg ha$^{-1}$ y$^{-1}$) (Fig. 5, Table A3), rather than by the dynamics of the compartment. Moreover, litter quality is a fundamentally important factor in determining the rates of C decomposition and stabilization under different management practices (Córdova et al., 2018). The two compared agricultural systems differ only in the plant species during the pasture phase of the R system, as the agricultural phase of this system is identical to the sequence of the CC system (Fig. 1). In addition, the mixture of species planted in the pasture phase include both high-quality litter (white clover, birdsfoot trefoil) and low-quality litter (tall fescue). This may explain the absence of differences in the dynamics of each unit mass of $C_{fast}$ between the treatments (parameter $k_1$). However, the quality of the C input was not explicitly considered in this model, and therefore, it is confounded with the effect of quantity. Hence, caution should be taken in assessing the causes and implications of the absence of differences between the $k_1$ outflow rate constants of both treatments.

The high outflow rates constants ($k_1$) of this compartment indicates that the fast pool is responsible for most of the C outflow in both systems (Fig. 5, Table A3) as evidenced by the radiocarbon signature of the incubations (Table 2). Overall, higher inputs in the R system coupled with similar dynamics of the labile pools explain ~18% of the difference in carbon between R and CC systems.

MAOM Persistence Mechanism

In contrast to the fast pool, slow pool dynamics was strongly influenced by the agricultural treatment. The C outflow rate constant from the slow pool ($k_2$) was 5.19 times higher in Period 1 (1964 – 1990) of the CC system compared to the R system, and 3.68 times higher in Period 2 (1991 – 2021) of the CC system compared to the R system (Fig. 3, Table A2), implying that

slow pool (~ MAOM) stability has been the main factor determining the observed differences between treatments. The difference between periods has been essentially caused by tillage intensity in Period 1, with also higher frequency in the CC system, which promoted aggregate destabilization and increased soil aeration, exposing the C contained in the MAOM fraction to microbial degradation (Rui et al., 2022; Spohn and Giani, 2011). This process was particularly relevant in the first phase of the experiment (before 1984), characterized by conventional tillage application, with the CC system having twice as many tillage interventions as the R system. This is consistent with the steeper slope of SOC loss observed until 1990 (Fig. 4b), determining a higher $k_2$ value in this period (Fig. 3, Table A2). However, it is striking that the $k_2$ parameter was found to be 3.68 times higher in the CC system from 1990 onwards, when reduced tillage technologies had already started to be implemented, evolving into a no-till system in 2009.

New approaches to SOC dynamics suggest that MAOM destabilization processes also depend on the availability of nutrients for the microbial biomass. Lower levels of labile C in systems with fewer inputs and lower POM stocks may lead to an increase in the destabilization of the MAOM pool, which would take on a role as a nutrient provider for microbial biomass (Daly et al., 2021). This may be one of the reasons why high relative rates of decomposition of the slow-cycling pool are sustained in particular in CC even after the incorporation of minimum tillage and no-till systems. Additionally, Hall et al. (2019) suggest that the alternation in crop rotations of N-rich inputs like soybeans and other crops with high C/N ratio litter like that produced by crops such as wheat, corn, or sorghum may enhance litter and stable SOC decomposition due to the priming effect generated by the high C/N litter, accelerated by the elevated growth of microbial biomass previously caused by the N-rich soybean litter. The frequency of soybean crop is twice as high in the CC system as in the R system (Fig. 1), so this process could explain part of the differences observed between treatments.

MAOM Accrual Mechanism

The C flow from the fast to the slow pool (stabilization) emerged as a second key factor in determining the SOC differences between the treatments (Fig. 5, Table A3). In the mathematical representation used in this study (Eq. 1, Eq. 2, Fig. 2), this process is governed by the coefficient α that represents the mass of C flowing to the slow-cycling pool (~ MAOM) per unit of C leaving the fast-cycling compartment (~ POM), and $k_1$ (the rate constant at which material leaves the fast pool). Although these coefficients did not differ significantly between the two agricultural systems, the higher C input rate in the R system (R: 4.94 Mg ha$^{-1}$ y$^{-1}$, CC: 2.87 Mg ha$^{-1}$ y$^{-1}$) results in a larger $C_{fast}$ stock, which in turn leads to a more intense C stabilization process, represented by a greater C flux to $C_{slow}$ (R: 0.078 Mg ha$^{-1}$ y$^{-1}$; CC: 0.052 Mg ha$^{-1}$ y$^{-1}$) (Fig. 5, Table A3).

Various studies agree that the processes of SOC stabilization in association with the mineral phase rely on a high level of microbial activity, with subsequent anabolic production of organic compounds (Cotrufo et al., 2015, 2013; Kallenbach et al., 2016; Kleber et al., 2011), that flow to stable compartments mediated by the formation of microbial necromass (Córdova et al., 2018; Zhu et al., 2020). The observed results are consistent with the intensification of these processes in the R system due to a greater C input rate.

As suggested by previous studies, different qualities in the C input between treatments could also be relevant in determining the stabilization processes previously discussed (Ma et al., 2018; Zhu et al., 2020; Córdova et al., 2018). However, the model used in this study does not explicitly represent the influence of input quality, which becomes confounded with the effects of quantities. Therefore, in this case, we can assert that the higher C stabilization process in integrated crop-pasture systems in R seems to be essentially explained by the quantity of C inputs, and there may be an effect of the quality that should be studied more deeply.

From a practical point of view, it is clear that the incorporation of no-tillage systems is fundamental for the conservation of old SOC. In turn, the inclusion of pasture phases with perennial species and constant C inputs to the soil has an effect of increasing the C stabilization rate, but primarily of reducing the losses of old SOC. This would essentially be related to the reduction of fallow periods between crops and the maintenance of relatively high stocks of labile C, which allows both the conservation of MAOM C that does not become the main substrate for microbial oxidation when there is a high quantity of C inputs (Daly et al., 2021), and the stimulation of MAOM C synthesis by microbial anabolism. It is important to interpret these results considering the management carried out in the R system, where the pastures are mowed and all biomass is allowed to return to the soil surface without grazing. This management was implemented in this experiment to facilitate operational aspects by avoiding the inclusion of direct grazing by animals. Since typical management of these agricultural systems producing beef and dairy would include either the presence of grazing animals in pasture fields, or the removal of biomass produced by hay harvest, future work should consider these effects on the dynamics of SOC, in an approach of compartmental model adjustment assisted by isotopic information.

C transit time and age

The model structure and its parameterization gave rise to emergent properties such as C transit time and age (Sierra et al., 2018b). The steady-state representation of the R system showed that, on average, each atom of C remains 11.5 years after entering the system (mean transit time) (Fig. 6d). However, the transit time distribution was strongly right-skewed, with half of the C entering this system remaining for only 1.09 years (median transit time). This value approximately matched the mean age of the fast-cycling pool (~ POM) (1.5 years) (Fig. 6b), making it clear that the system's outputs are predominantly explained by a labile pool decomposing at high rates and very low outputs of old C from the slow-cycling pool (~ MAOM) (mean age 632 years) (Fig. 6c), which heavily skews the C outflow age distribution to the right (Fig. 6d).

Regarding the CC system, the age distribution of the fast pool did not differ from that of the R system (Fig. 6b), determining that half of the C entering this system remains for similar periods (median transit time 1.15 years). However, the mean transit time in CC system reached lower values than in R system, though these differences were not statistically significant (CC system: 6.2 years; R system: 11.5 years) (Fig. 6d). This is explained by the different dynamics of the slow pool, which, because of higher output rates in CC system, tends to become younger as the new C entering this compartment is less diluted in a large

stock of old C. This is consistent with the lower average ages of the slow pool (Fig. 6c) and the bulk soil in this system (Fig. 6a), and with the even more right-skewed distributions observed for CC system in steady-state condition compared to the R system. Analyzing the transit time results for both systems, we found them close to those observed at a long-term experiment in New Zealand with a similar climate (Stoner et al., 2021). Additionally, it is important to highlight the difference in mean transit time between treatments (R: 11.5 years; CC: 6.2 years), even though the left end of their distribution, shown in figure 6d, is very similar. The R System has a slow-cycling pool much larger than the CC at equilibrium and with a much older mean age (Fig. 6c), which determines that the transit time distributions of both systems are different at their right end (determined by the contribution of the slow pool) but very similar at their left end (determined by the contribution of the fast pool). This difference explains why the mean transit times are disparate even though the distributions look nearly identical.

Focusing on the age structure of C in the R system, we found that 21.4% of its C (12.10 Mg ha$^{-1}$) was incorporated during the period of the experiment (59 years), of which more than half (7.58 Mg ha$^{-1}$) consists of fast-cycling C (~ POM), a pool that in its entirety is younger than the age of the experiment. The remainder of the total renewed C corresponds to slow-cycling C (~ MAOM), and only 8.8% of the C from this pool entered the system after the year 1963; therefore, most of this pool was already present in the soil before the experiment's installation. Analyzing the age structure of slow pool specifically, 85.4% is older than 100 years, 44.9% is older than 500 years, and 19.9% is older than 1000 years. Conducting the same analysis for the CC system would be meaningless as it has not yet reached a steady state, a necessary assumption for calculating these properties (Manzoni et al., 2009; Sierra et al., 2021). For this reason, the distributions presented in Fig. 6 for the CC system do not represent the current age structures but rather those that will occur once the system reaches steady state if its current dynamics are projected forward. However, it is clear that the CC system induces a "renewal" effect by significantly losing very old C from the $C_{slow}$ pool, which progressively becomes more diluted with the inputs to this compartment through transfers from the $C_{fast}$ pool.

Finally, it is important to note that the model structure used in this work assumes that all inputs enter the system through the $C_{fast}$ pool, although there is scientific evidence to support alternative structures (Cotrufo et al., 2013, 2015). Nevertheless, we understand that this simplification does not have significant implications on our results and their interpretation. A large difference in the isotopic signature of $\Delta^{14}C$ between very negative bulk C and modern $CO_2$ efflux was observed, which implies that most of the C inputs must cycle through a fast compartment (~ POM) and only a minor proportion may flow through a slow pool (~ MAOM). Therefore, to keep the $\Delta^{14}C$ signature of bulk C consistent with the measured data, a direct C input to the slow pool was considered negligible as it should occur at a sufficiently low rate not to "rejuvenate" that pool. Moreover, considering an alternative model structure with direct inputs to the slow pool, would have required additional information for the adjustment of an additional parameter. Future work with enough availability of isotopic information should test new hypotheses regarding alternative model structures in these types of agroecosystems.

## 5 Conclusions

Based on a modeling process that uses data from a 60-year monitoring experiment, as well as radiocarbon measurements from soil and incubations, we conclude that the greater SOC stock in integrated crop-pasture systems compared to continuous annual grain systems is primarily caused by the preservation of old carbon, because of lower outflow rates of the slow-cycling pool. The persistence of slow C (~ MAOM) in the integrated system leads to a highly old age distribution, with only 8.8% of the carbon in this pool having entered the system since the experiment's initiation in 1963. Another important factor in determining

the current difference in carbon stocks between the systems is the higher flow of carbon from the fast to the slow pool (stabilization). This process may occur to a greater extent in the integrated system due to higher carbon inputs, which may result in a larger stock of labile C and consequently greater microbial activity. Although no differences were observed in the model parameters ($k_1$ and $\alpha$) that determine the C flow to the slow-cycling pool per unit of C entering the system, it was not possible to establish a mechanistic connection with the quality of the C input, given that this factor was not explicitly modeled.

Regarding the implications of these results on the management of agricultural systems, it is important to maintain a high rate of well-distributed C inputs over time to feed the microbial community, preventing the destabilization of old C. Therefore, the central importance of reducing fallow periods and increasing system productivity to raise C inputs becomes relevant, not only to enhance the stabilization processes of new C but mainly as a mechanism to prevent the loss of old C.

Future work should consider the effects of soil erosion on radiocarbon dynamics and its implications in model's adjustment and interpretation. Additionally, the evaluation of integrated crop-pasture systems established on already-degraded soils that are on pathways to recovering carbon stocks should also be considered.

## Appendices

Appendix A

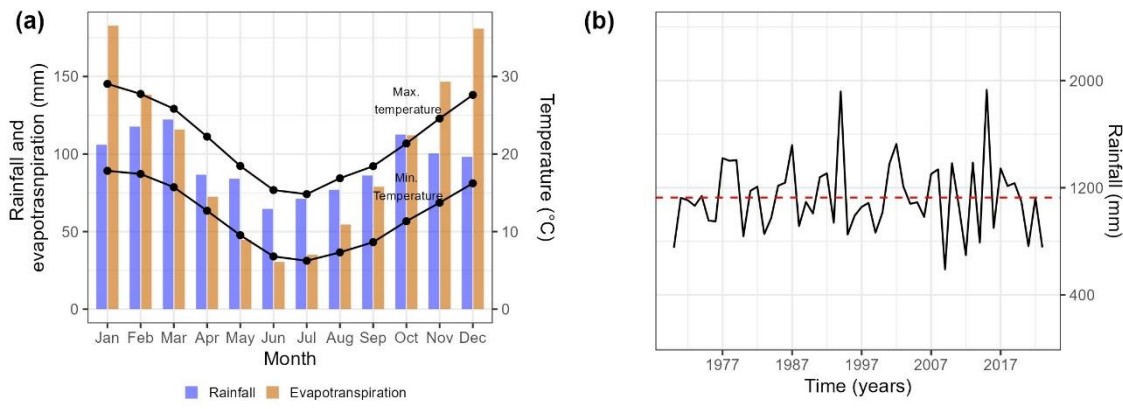

**Figure A1.** Monthly distribution of mean accumulated rainfall and evapotranspiration (left Y axis, bars) and monthly mean maximum and minimum temperature (right Y axis, lines) (a); time series of annual accumulated rainfall (mm) (dotted red line corresponds to the time series average) (b). The information derives from a 53-year time series (1969 – 2022) provided by INIA's meteorological stations (INIA GRAS, 2023).

**Table A1.** Soil characteristics of La Estanzuela LTE

| Horizon | $A_p$ | $B_{t1}$ | $B_{t2}$ | $B_{t3}$ |
|---|---|---|---|---|
| Depth (cm) | 0-30 | 30-42 | 42-72 | 72-97 |
| Clay (g kg$^{-1}$) | 287 | 472 | 500 | 466 |
| Silt (g kg$^{-1}$) | 637 | 466 | 461 | 486 |
| Sand (g kg$^{-1}$) | 76 | 62 | 39 | 48 |
| pH ($H_2O$) | 5.6 | 6.1 | 6.6 | 6.8 |
| pH (KCl) | 5.6 | 5.6 | 5.2 | 5.5 |
| SOC (g kg$^{-1}$) | 20.8 | 7.8 | 9.5 | 1.3 |
| $N_{tot}$ (g kg$^{-1}$) | 1.7 | 0.8 | 0.7 | 0.4 |
| Ca (cmol$_c$ kg$^{-1}$) | 14.0 | 19.0 | 20.0 | 22.3 |
| Mg (cmol$_c$ kg$^{-1}$) | 2.0 | 2.7 | 3.0 | 3.3 |
| K (cmol$_c$ kg$^{-1}$) | 0.8 | 0.8 | 0.9 | 1.0 |
| Na (cmol$_c$ kg$^{-1}$) | 0.3 | 0.4 | 0.5 | 0.5 |
| $CEC_e$ (cmol$_c$ kg$^{-1}$) | 17.1 | 22.9 | 24.4 | 27.1 |
| $CEC_{pH7}$ (cmol$_c$ kg$^{-1}$) | 20.6 | 25.5 | 26.6 | 28.4 |
| Ef. Base Sat. (%) | 100 | 100 | 100 | 100 |

Note: CECe: efective Cation Exchange Capacity; Ef. Base Sat: (Total Bases/CECe) * 100. Information obtained from a soil mapping conducted by the Uruguayan Ministry of Agriculture in 1985.

**Table A2.** Summary information of the model parameters populations for each agricultural system (R and CC) obtained through the Bayesian fitting procedure (MCMC)

| | R | | | CC | | | | |
|---|---|---|---|---|---|---|---|---|
| | k1 | k2 | α | k1 | k2 (P1) | k2 (P2) | α (P1) | α (P2) |
| Mean | 0.652 | 1.59E-03 | 1.58E-02 | 0.633 | 8.25E-03 | 5.85E-03 | 1.69E-02 | 2.46E-02 |
| SD | 0.018 | 2.35E-05 | 2.65E-04 | 0.099 | 1.48E-03 | 1.76E-03 | 1.77E-02 | 2.15E-02 |
| Min | 0.596 | 1.45E-03 | 1.44E-02 | 0.406 | 5.06E-03 | 2.00E-03 | 1.43E-06 | 2.45E-06 |
| Max | 0.730 | 1.72E-03 | 1.71E-02 | 1.047 | 1.43E-02 | 1.14E-02 | 8.00E-02 | 7.98E-02 |
| Q1 | 0.639 | 1.57E-03 | 1.56E-02 | 0.558 | 7.23E-03 | 4.58E-03 | 4.04E-03 | 6.24E-03 |
| Q2 | 0.651 | 1.58E-03 | 1.57E-02 | 0.619 | 8.05E-03 | 5.46E-03 | 1.04E-02 | 1.82E-02 |
| Q3 | 0.664 | 1.60E-03 | 1.59E-02 | 0.688 | 9.11E-03 | 6.99E-03 | 2.44E-02 | 3.87E-02 |

| | | | | | | | | |
|---|---|---|---|---|---|---|---|---|
| LL | 0.618 | 1.54E-03 | 1.53E-02 | 0.477 | 5.83E-03 | 3.30E-03 | 4.74E-04 | 5.90E-04 |
| UL | 0.690 | 1.64E-03 | 1.63E-02 | 0.871 | 1.17E-02 | 9.83E-03 | 6.69E-02 | 7.28E-02 |

Note: SD: standard deviation; Q1: first quartile; Q2: median; Q3: third quartile; LL: lower limit of 95% credible interval; UL: upper limit of 95% credible interval; k1: POM pool decomposition rate; k2: MAOM pool decomposition rate; α: transfer coefficient between pools; P1: Period 1 of CC treatment; P2: Period 2 of CC treatment.

**Table A3.** Main C flux rates derived from the compartmental model fitted to each agricultural system (for steady state condition for R system and for 2021 in CC system)

| | R | CC |
|---|---|---|
| C input rate (Mg ha$^{-1}$ y$^{-1}$) | 4.94 (1.9) | 2.87 (1.1) |
| Total C output rate (Mg ha$^{-1}$ y$^{-1}$) | 4.94 (2.17E-05) | 3.00 (2.4E-02) |
| $C_{fast}$ output rate (Mg ha$^{-1}$ y$^{-1}$) | 4.86 (1.3E-03) | 2.81 (6.3E-02) |
| $C_{slow}$ output rate (Mg ha$^{-1}$ y$^{-1}$) | 0.078 (1.3E-03) | 0.20 (6.2E-02) |
| $C_{fast}$-to-$C_{slow}$ transfer rate (Mg ha$^{-1}$ y$^{-1}$) | 0.078 (1.4E-03) | 0.052 (6.2E-02) |
| Stabilization efficiency (%) | 1.57 (0.025) | 1.85 (2.2) |

Note: The value in parentheses indicates the standard deviation of the mean. Stabilization efficiency: $C_{fast}$-to-$C_{slow}$ transfer rate / C input rate

**Funding statement**

Funding for this work was provided by the cooperation project "Understanding how land management alters C and N cycling in Uruguayan agro-ecosystems" (ANII MPI_ID_2018_1_1008457) between the Instituto Nacional de Investigación Agropecuaria (INIA – Uruguay), the Agencia Nacional de Investigación e Innovación (ANII - Uruguay), and the Max Planck Institute for Biogeochemistry (MPI-BGC - Germany), under the collaboration agreement between INIA- Uruguay and the Max Planck Society (MPG -Germany). The first author received a fellowship from the Comisión Académica de Posgrado (CAP) of the Universidad de la República (Uruguay) and an internship grant from the Comisión Sectorial de Investigación Científica (CSIC) of the Universidad de la República (Uruguay), which were fundamental for the development of this work.

**Code and data availability**

All code for reproducing the results is available at this repository: https://doi.org/10.5281/zenodo.11116986.

## Author contribution

Conceptualization: MGS, CAS, MVP. Data Curation: JAQ, MVP. Formal analysis: MGS. Funding acquisition: MVP. Investigation: MGS. Supervision: MVP, CAS, JAQ, WEB, ST. Writing – original draft preparation: MGS. Writing – review and editing: MGS, MVP, CAS, JAQ, WEB, ST. All authors have read and agreed to the published version of the manuscript.

## Competing interests

The authors declare that they have no conflict of interest.

## Acknowledgements

The authors are grateful to Emiliano Barolín, Eduardo H. Vergara and Gualberto Soulier for their support during the sampling campaigns, as well as to the historical staff of INIA for their contribution to the development and maintenance of the long-term experiment. We would also like to thank Iris Kuhlmann, Axel Steinhof, and Andrés Tangarife-Escobar for their support

in the laboratory activities at MPI-BGC that contributed to the data acquisition for this work. We are also grateful to Lucía Salvo for her help with the SOC fractionation technique and to the two anonymous reviewers whose suggestions greatly improved the final version of this work.

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
