# Peer review of "High capacity of integrated crop-pasture systems to preserve old soil carbon evaluated in a 60-year-old experiment"

_EGUsphere, 2023_

## Author Comment (AC3)

Dear Anonymous Reviewer No. 1,

I post a new comment to correct an error in my previous response, for which I apologize. In your comment:

L189: Were these C values adjusted to the KrCr2O7 method values as was done with the LECO C values?

I responded that yes, all values had been adjusted, but this was a mistake, as the latest version of the code that generates the results did not contain the correction for the coefficient of 0.81. Once I effectively applied this correction it became clear that the values of MAOM recovered in the fractionation were low, which led us to modify the data used in the following way:

- The measured POM data was corrected by a coefficient of 0.81.

- The MAOM fraction was calculated as the difference between soil bulk C and POM fraction. This result was then corrected by 0.81.

All the necessary information for these calculations is available in the file DATA.xlsx of the supplementary material, sheet 't_tests':

- C_stock (column F) is the bulk C stock for each plot, and C_stock_POM is the POM C for each plot. This data was corrected by multiplying it by a coefficient of 0.81, and the C stock in MAOM was calculated by difference.

These recalculated POM-MAOM ratios were used then to run the modeling again. This did not modify the results and conclusions of the work.

I apologize again for the mistake in my previous response.

Regards,

Maximiliano González

---

## Author Response (AR1)

Author's response

We want to thank the editor for the opportunity to resubmit the article for your consideration with the corrections made. This document details all the changes made to the manuscript. First, the responses to each of the editor's comments appear, followed by the responses to reviewer 1 and then the responses to reviewer 2.

1. EDITOR COMMENTS

I appreciate the inclusion of the concepts of radiocarbon and modeling included in the introduction and am supportive of the authors' decision to keep them in the introduction while still addressing Reviewer #1's request for additional introductory text on POM and MAOM. Please, however, make very clear that POM and MAOM are not perfect embodiments of the conceptualizations of the fast and slow pools (i.e. they are not completely homogenous or unique fractions, and each may include fast- and slow-cycling components).

This text was added at the end of the introductory text on POM and MAOM (L 48):

"However, although they constitute a good proxy for characterizing compartments with different kinetics (Poeplau et al., 2018), they are not completely homogeneous compartments, and POM may include some proportion of slow cycling C, while MAOM may include some proportion of fast C."

The text suggested by the authors to address Reviewer #1's request for lines 44-45 does not quite address the issue, in my opinion. The first sentence is still a bit misleading because of how it is worded, while the second sentence ignores the role that losses might play in increasing SOC stocks (i.e., inputs do not necessarily need to increase for SOC stocks to increase). I therefore suggest further revision, possibly something like the following:
"Low carbon inputs and intensive tillage have been identified as some of the main causes of soil deterioration and losses of soil organic carbon from agricultural systems (Rui et al., 2022). Increasing SOC stocks requires either increased carbon inputs (e.g. King and Blesh (2018)) without compensatory SOC losses, or decreased SOC losses relative to inputs."

Changed according to editor's suggestion.

In the revised text addressing Reviewer #1's comment for lines 92-96, please replace "hypothetic" with either "hypothesized" or "hypothetical".

Changed according to editor's suggestion.

Please modify the text to indicate the number of samples included in the laboratory validation process, as raised by Reviewer #2 (originally line 172).

Changed according to editor's suggestion.

In response to Reviewer #2's final comment (originally line 503-505), I believe the modified text is still too speculative given that microbial activity was not directly measured and a causal link to inputs was not directly tested, therefore request that the sentence be further modified to indicate these are potential mechanisms, (e.g. change "occurs" to "may occur", and "result" to "may result"). If, however the authors believe they can more clearly link a mechanism to their results, I would welcome that approach here instead so long as it is not speculative.

The text was modified according to the editor's suggestion to indicate that potential mechanisms are being proposed.

Finally, I would appreciate some text in the discussion focused on the implications of the assumptions represented in the model structure for the results. For example, if the model did not assume the fresh litter enters the fast pool before the slow pool, sequentially, but allowed for some litter to directly form slow-cycling SOC, might that have potential to change the results or interpretation? I realize the authors can not identify and explain every possible model structure here, and don't mean to suggest that this must be exhaustive in any way. However, given that there is evidence in the literature to support alternate model structures, an acknowledgement that some of the assumptions inherent in the structure may not align with current understanding of SOM formation, and some discussion of how the assumptions may or may not influence the results (or justification of the choices made), would strengthen the manuscript.

Text was added to the end of the discussion section in response to the editor's suggestion (L.563):

"Finally, it is important to note that the model structure used in this work assumes that all inputs enter the system through the $C_{fast}$ pool, although there is scientific evidence to support alternative structures (Cotrufo et al., 2015, 2013). Nevertheless, we understand that this simplification does not have

significant implications on our results and their interpretation. A large difference in the isotopic signature of $\Delta^{14}C$ between very negative bulk C and modern $CO_2$ efflux was observed, which implies that most of the C inputs must cycle through a fast compartment (~ POM) and only a minor proportion may flow through a slow pool (~ MAOM). Therefore, to keep the $\Delta^{14}C$ signature of bulk C consistent with the measured data, a direct C input to the slow pool was considered negligible as it should occur at a sufficiently low rate not to "rejuvenate" that pool. Moreover, considering an alternative model structure with direct inputs to the slow pool, would have required additional information for the adjustment of an additional parameter. Future work with enough availability of isotopic information should test new hypotheses regarding alternative model structures in these types of agroecosystems."

Additional comments:

- The description of the botanical species sown in the pasture phase of R system was taken from a previous description of the experiment. However, professionals in charge of the experiment informed us that the pasture mixture does not contain red clover. It is a mix of white clover (Trifolium repens L.), birdsfoot trefoil (Lotus corniculatus L.), and tall fescue (Festuca arundinacea Schreb.). Red clover was removed from the description.

- The last data point (year 2021) for SOC time series in R system (Fig. 4a) was incorrectly calculated. It is corrected in the new figure.

- POM-MAOM data was recalculated as described in the second response to reviewer 1.

- C inputs for R system were recalculated according to reviewer 1's suggestions. A review of the calculations initially made for the response to reviewer 1 resulted in a slight modification of the value stated in that document for R system from 4.89 Mg ha-1y-1 to 4.94 Mg ha-1y-1.

- The bulk densities used for the calculation of the SOC time series were modified. Previously: 1.24 ± 0.03 Mg m-3 (R system), 1.35 ± 0.04 Mg m-3 (CC system). Modified values: 1.28 ± 0.02 Mg m-3 (R system), CC: 1.38 ± 0.03 Mg m-3 (CC system). The initially considered bulk densities corresponded to the 0-10 cm layer and not the 0-20 cm layer; the new values correspond to the average bulk density of the 0-20 cm layer.

All these modifications were considered to reparametrize the model for each system. Other than updating specific numbers for slight changes as can be observed in the new version of the manuscript, this reparameterization did not imply any changes in the results discussed.

2. REVIEWER 1:

I have read "High capacity of integrated crop-pasture systems to preserve old stable carbon evaluated in a 60-year-old experiment." The manuscript describes a study that models the POM and MAOM soil dynamics in continuous grain and rotational agricultural systems by combining long-term field SOC measurements with intermittent bulk sand respired soil $CO_2$ radiocarbon data. The model indicated that SOC loss in the continuous grain system was primarily due to loss of MAOM, which was attributed to a higher outflow rate (decomposition) of the MAOM pool compared to the rotational system. The POM pool was also smaller in the continuous system, but proportional outflows (decomposition and transfer to MAOM) were similar to the rotational system. The authors conclude that the high C input into the POM pool in the rotational system maintains the MAOM pool, whereas the lower C input into the POM pool in the continuous system facilitates MAOM loss.

This manuscript is generally well-written, and the topic is of high interest. Long-term SOC datasets are very valuable, particularly when combined with radiocarbon data. To improve the manuscript, I have some comments that the authors may wish to consider.

We are very grateful for the valuable comments raised by the reviewer. Below, we present the answers to each of the points made. In many cases, these responses led to modifications in the original manuscript, so we outline the lines (corresponding to a new version of the manuscript) where these modifications were introduced.

General comments:

The introduction could be improved to better set up the ideas presented in the rest of the manuscript. Specifically, the concepts of POM and MAOM are not mentioned until the objectives at the end of the discussion, yet the POM and MAOM pool dynamics are the centerpiece of this manuscript. The concepts of POM and MAOM should thus be a central part of the introduction. In contrast, methodological details about radiocarbon (L61-63) and model specification (L73-78), which are tools used in the study, can be briefly summarized or omitted from the introduction and placed in the methods.

We agree with the reviewer regarding the need to develop the concepts of POM and MAOM in the introduction. These have been added in a new paragraph in this section (L.38-50). We also believe that the use of radiocarbon, as well as the application of compartmental models and their interpretation, are also central to the development of this work. Therefore, we may rather keep the main concepts relevant to these tools in the introduction, while attempting to partially summarize them at the same time.

Several times throughout the discussion the authors mention that the difference between the rotational and continuous grain system is primarily caused by differences in C input quantity, not quality. However, the rotational system contains many legumes, which differ drastically in quality compared to grain. The authors surmise that litter quality is not important because the POM pools of both the rotational and continuous system had similar outflow proportions. Given that in this case litter quantity is confounded with litter quality, and that the model did not explicitly account for litter quality, it is very speculative to make statements about how litter quality affected the measured pools. I recommend that the authors reduce their speculations about this and in all cases make it clear that it is highly speculative and warrants more research.

The reviewer makes a good point. The discussion has been modified accordingly, to qualify the claims regarding the effect of input quality and to highlight the need for further research. Specific answers to the detailed comments made by the reviewer later in this document are outlined.

A limitation of this study is that only the top 20 cm was measured and modeled, and the entire 20 cm was treated as a homogenous entity. Especially considering that these systems are now minimal or no-till, the top few centimeters of soil receive most of the aboveground inputs, and therefore they may be quite different than the soil below it. Moreover, root inputs are certainly present below 20 cm, but the model assumes that they are all within the top 20 cm. These limitations should be mentioned in the discussion.

Regarding the modeling of the 0-20 cm layer as a homogeneous entity, this text was added in L.434:

"Another point that must be considered in interpreting the results of this work is that the available information only allowed us to consider the 0-20 cm layer as a homogeneous entity, both in terms of C inputs and SOC dynamics. This aspect could become relevant from 2009 onwards, when no-till system was adopted. Obtaining information with much more intensive sampling stratification schemes than those employed in this study could enhance the historical data of this experiment, enabling it to address new hypotheses in this direction."

Regarding the consideration of only the proportion of root inputs in the 0-20 cm layer, it was indeed considered in the construction of the C inputs, but this was not properly detailed in the methodological description of the original manuscript. This explanation has been added in L.210:

"The proportion of belowground C inputs corresponding to the 0-20 cm layer was calculated by considering a coefficient of 0.72, obtained from the ratio

between C POM (0-20 cm) and C POM (0-80 cm) for the site (data not shown) and assuming that the vertical distribution of roots correlates positively with the vertical distribution of C POM."

An important caveat with the "crop-pasture rotational system" was that the pasture phase was not grazed or harvested. Instead, the pasture phase was mowed, and all biomass was allowed to return to the soil surface. While this does not detract from the mechanistic understanding gained from the research, it is important to specify that the results are likely a "best case scenario" in terms of C inputs and thus soil C storage. That is, if the pasture were a true agricultural system that was grazed or harvested for hay, then the total C inputs would likely have been much lower than the current mowed system. This should be mentioned in the discussion as to properly place the results into the context of realistic agricultural system management.

To address the reviewer's comments, the following text has been added at line L.526:

"It is important to interpret these results considering the management carried out in the R system, where the pastures are mowed and all biomass is allowed to return to the soil surface without grazing."

Some of the figures and tables seem to be redundant or supplemental in nature, while others could be improved clarity. More details are provided in the specific comments.

This point was covered in response to the specific comments made later in this document.

Specific comments:

L12-13: This would be a good place to qualify that these differences between pasture and continuous cropping systems (in this study and others) are mostly seen in surface soils (e.g., 0-20 or 0-30 cm).

Text modified in lines L.12-13 to clarify this point.

"Integrated crop-pasture rotational systems can store larger amounts of soil organic carbon (SOC) stocks in the topsoil (0-20 cm) than continuous grain cropping."

L14-16: This would be more accurate if it read "We analyzed the temporal changes of 0-20 cm SOC stocks …"

Text modified in lines L.15 to clarify this point.

"We analyzed the temporal changes of 0-20 cm SOC stocks in two agricultural treatments of different intensity (continuous cropping annual grain and crop-pasture rotational systems) in a 60-year experiment in Colonia, Uruguay."

L22-24: This sentence is somewhat unclear. To make the case that loss of MAOM-C was the important factor, it may be good to compare the age of pasture MAOM (~ 600 years) with the age of continuous crop MAOM (~ 200 years).

Text modified in lines L.23-25 to try to clarify the sentence.

"The avoidance of old C losses in the integrated crop-pasture rotational system resulted in a mean age of the slow-cycling pool (~ MAOM) over 600 years, with only 8.8% of the C in this compartment incorporated during the experiment period (after 1963) and more than 85% older than 100 years old in this agricultural system."

Anyway, MAOM-C (~ slow pool) dynamics is the main difference between treatments since the k2 adjusted for the CC system was between 3.68 to 5.19 times higher than in the R system. The low decomposition rate of the MAOM pool in the R system is what determines the minimal incorporation of modern C, which is the point raised by the sentence mentioned by the reviewer in this case (only 8.8% of the mass of this compartment entered after 1964 in the R system, the rest is older).

Following a suggestion made by reviewer 2, we also modified the terms POM and MAOM to now refer to 'fast' and 'slow' conceptual pools, whose sizes were estimated based on the abundance of POM and MAOM in 2021.

L30: Should "debate" be "discussion?"

Changed.

L31-35: "On one hand/on the other hand" implies two contrasting statements, but these statements are not contrasting. Consider using different phrasing.

Text was changed to improve phrasing.

"Firstly, it is the primary indicator of soil quality, because of its direct relationship with the physical, chemical, and biological properties that

determine soil fertility and productivity (Reeves, 1997). Additionally, soils contain approximately two times more C than the atmosphere (Jobbágy and Jackson, 2000; Janowiak et al., 2017), and therefore, slight increases in their storage have the potential to reduce atmospheric CO2 levels and contribute to the fight against climate change (Fargione et al., 2018)."

L31-33: Please provide a citation for this statement.

Citation added: (Reeves, 1997).

L44-45: It is a bit misleading and uninformative to suggest that "continuous monoculture" is "responsible for emissions of large amounts of C..." First, there is previous work to suggest that diversifying crops/rotations will likely only be effective at increasing SOC if the diversified system results in increased C inputs (King & Blesh 2017). Second, "continuous monoculture" describes a system rather than a mechanism; that is, it would be much more informative to list the mechanism by which a continuous monoculture could reduce SOC, for example, reduced inputs and increased tillage.

Text changed in L.59-61 to clarify this point.

"Low carbon inputs and intensive tillage have been identified as some of the main causes of soil deterioration and losses of soil organic carbon from agricultural systems (Rui et al., 2022). Increasing SOC stocks requires either increased carbon inputs (e.g. King and Blesh (2018)) without compensatory SOC losses, or decreased SOC losses relative to inputs."

L44-55: These ideas are really the core of the manuscript, and I think this section warrants more length and detail. In particular, this is where POM and MAOM should be introduced as the manifestations of the "labile" and "stable" pools. There is also previous work in this area that can be explored here, such as King et al. (2020), Prairie et al. (2023), and references therein, that evaluate litter C input, quality, and SOC stabilization.

A new paragraph was added starting from line 38 to give more emphasis in the introduction to the SOC fractions that are central in the work:

"SOC is a heterogeneous mixture of different components that decompose at different rates (Kögel-Knabner et al., 2008), and modeling SOC dynamics as a single pool overestimates the system's response on time scales of decades to centuries (Trumbore, 2009). In this context, the separation of SOC pools with different kinetics is fundamental for the accurate representation of their dynamics (Lavallee et al., 2020). The separation into particulate organic matter

(POM) and mineral associated organic matter (MAOM) (Cambardella and Elliott, 1992) is one of the fractionation techniques that has proven to be highly effective. POM is composed of low-density materials, with little microbial processing and chemical characteristics close to the plant input material, while MAOM is a fraction protected from decomposition through association with the mineral phase, where individual molecules or small fragments of organic matter predominate with a greater contribution of microbial-derived compounds (Lavallee et al., 2020). Because of the different stabilization processes that characterize each of these fractions, on average MAOM tends to have lower decomposition rates (longer persistence) than POM (Poeplau et al., 2018; Trumbore and Zheng, 1996; Heckman et al., 2022). However, although they constitute a good proxy for characterizing compartments with different kinetics (Poeplau et al., 2018), they are not completely homogeneous compartments, and POM may include some proportion of slow cycling C, while MAOM may include some proportion of fast C.Minimal changes were also made on lines 60 and 68 -70 with the same objective."

L45-47: Please provide a citation for this statement. Please also clarify whether these findings pertain to the entire soil profile, shallow depths, lower depths, etc.

The citation for this statement is the same as for the following one. The wording has been modified for clarity. Now in L.61 onwards:

"Various management practices such as reduced tillage, crop diversification, and application of amendments, have proven to be effective in increasing the poorly transformed particulate fractions of topsoil (0-30 cm) organic matter, but their effectiveness in generating more persistent SOC in association with the mineral phase has been debated (Ogle et al., 2012; Rui et al., 2022)."

L60-61: The 14C isotopic signature also reflects the 14C signature of the plant inputs.

Sentence modified in lines 78-80 of the new document.

"In the particular case of SOC, which constantly receives new C inputs via photosynthesis and loses it through decomposition, the $^{14}$C isotopic signature of SOC reflects the $^{14}$C signature of the C input, the decomposition rate, and the radioactive decay rate of this isotope (Trumbore, 2000)."

L61-80: Radiocarbon and differential equation models are some of the tools used to study SOC (POM and MAOM) dynamics, but the SOC dynamics, not the tools, are the primary subject of this study. As such, I suggest only alluding to

the tools in the introduction and then explaining them in more detail within the methods section.

We believe that for this work, the study of radiocarbon and the characteristics of the models used are as central to the determination of the results as the SOC fractionation technique itself. The characterization of the dynamics of the analyzed systems, summarized in the parameters of the adjusted model (k1, k2, alpha), as well as the characterization of the age structure and transit time of C, arise from the use of radiocarbon data and are influenced by the assumptions considered in the structure of the model used. Therefore, we find the emphasis placed on the description of these two points to be relevant. However, we agree with the reviewer that it was necessary to go deeper into aspects related to the POM-MAOM fractionation technique and the characteristics of the fractions, which was included in the new paragraph starting from line 38. Nonetheless, some adjustments were made to the text of the paragraphs between lines 75-93 (new document) to summarize the concepts related to radiocarbon and SOC modeling. For example, the definitions of system age and transit time were moved to the methodology section.

L85: "Temporal changes" is a more explicit way to describe the "evolution." I suggest removing "evolution" and replacing it with "temporal changes" or something similar.

Changed according to the reviewer's suggestion.

L92-96: "Alternative hypotheses" implies that the hypotheses are mutually exclusive, but in this case, more than one hypothesis could be true. The "hypotheses" outlined are more akin to theories, where each theory could be false or true (null and alternative hypothesis, respectively). I suggest rewording this section to better describe these ideas.

We changed the text following the reviewer's suggestion to avoid referring to mutually exclusive hypotheses:

"To address the objective, the following possible mechanisms explaining the higher C storage in integrated crop-pasture rotational systems compared to intensive agriculture were hypothesized: 1) large input rates that promote SOC stabilization processes that support a high SOC stock (hypothetical MAOM accrual mechanism); 2) large input rates that promote the accumulation of large stocks of poorly stabilized particulate C (hypothetical POM accrual mechanism); 3) high persistence of very old SOC linked to low oxidation rates of passive SOC pools (hypothetical MAOM persistence mechanism); 4) a combination of the previous processes."

L106: This states that "rainfall is highly variable among years, but it does not show a long-term trend (Fig. 1b)," yet Fig. 1b shows 3-month average rainfall, and thus neither of these points can be visualized in the graph. The graph should be changed to annual rainfall so that the reader can evaluate interannual variability and trends if that is the goal here.

The graph in Figure 1b was changed according to the reviewer's suggestion.

[Figure]

**Figure A1.** Monthly distribution of mean accumulated rainfall and evapotranspiration (left Y axis, bars) and monthly mean maximum and minimum temperature (right Y axis, lines) (a); time series of annual accumulated rainfall (mm) (dotted red line corresponds to the time series average) (b). The information derives from a 53-year time series (1969 – 2022) provided by INIA's meteorological stations (INIA GRAS, 2023).

L109-114: Figure 1 is ancillary in nature and therefore I suggest that it can be placed in supplemental information.

Figure 1 was moved to Appendix A following the reviewer's suggestion. Now Figure A1 in new manuscript.

L123-124: Considering that this study only reports SOC dynamics to 20 cm deep, it seems unnecessary to report soil characteristics for deeper depths in Table 1. Moreover, most of the data presented in Table 1 are mostly ancillary and thus may not warrant inclusion in the main text. I suggest moving Table 1 to supplemental and summarizing some key components such as SOC, N, clay, and silt for the top 30 cm in the text on L120. E.g., "In 1985, a soil survey at the site reported SOC mass fraction of 20.8 g kg-1, N mass fraction of 1.7 g kg-1, 28.7% clay, and 63.7% silt (Table S1) for the 0-30 cm depth (Table S1)."

Table 1 was moved to Appendix A (Table A1), and the summary text suggested by the reviewer was added (lines 126-127):

"In 1985, a soil survey at the site reported SOC mass fraction of 20.8 g kg$^{-1}$, N mass fraction of 1.7 g kg$^{-1}$, 28.7% clay, and 63.7% silt for the 0-30 cm depth (Table A1)."

L135: Should "cropping sequence" be "crop rotation sequence?"

Text changed.

L139: It is unclear which treatments/rotation phases were tilled. In particular, did the grain phase of the rotational pastures receive tillage. Perhaps this could be explained in the next paragraph, around L151 and could be integrated into Figure 2.

Yes, the crop phase of the rotation system received the same tillage interventions as the continuous cropping system. The wording from line 143 of the new document has been modified to try to clarify this point.

"In the first 20 years of the experiment, soil preparation was carried out with conventional tillage (moldboard and disk plow) for the establishment of all crops and pastures in all treatments. Starting in 1983, the use of a chisel plow was gradually adopted, and pastures were sown in association with the last crop in the integrated crop-pasture treatments to avoid tilling intervention. From 2009 onwards, no-till farming was adopted in all treatments, eliminating the mechanical operations of soil preparation."

L145-146: Is there a more descriptive phrase than "continuous agriculture" to describe the CC treatment? For example, a "continuous annual grain system?"

Changed to continuous annual grain system throughout the document.

L150-151: For readers who may be unfamiliar with pastures, it may be worth noting here that three of the four pasture species are legumes, which are high in protein/N.

The words "legumes" and "grasses" were inserted for clarification in lines 155. The botanical composition of the pasture is relevant information regarding the chemical characteristics of the input.

"The R system includes three years of the crop sequence from CC, rotating with three years of a perennial pasture that consists of a mixture of legumes: white clover (Trifolium repens L.), birdsfoot trefoil (Lotus corniculatus L.), and grasses: tall fescue (Festuca arundinacea Schreb.)."

L156: While I understand the reasoning for mowing the pastures, the differences between a mowed pasture versus a grazed pasture should be noted here. Importantly, the amount of C returned to a mowed pasture is likely

much higher than a grazed pasture, as the metabolism of the grazer would consume much of the C. Moreover, there are likely differences between plant vs. manure C inputs.

A clarification in this regard is added to the discussion (L.526) to contextualize the scope of the results of this work.

"It is important to interpret these results considering the management carried out in the R system, where the pastures are mowed and all biomass is allowed to return to the soil surface without grazing."

L159 (Fig. 2): It is unclear to me why the far-right boxes (end of the rotation) are colored beige and contain the same crop as the treatment at the far left (beginning of the rotation). Is this meant to imply that the rotation repeats? If so, I suggest simply writing the word "repeat" on the right side of the rotation, otherwise it suggests that there are two sequential years of the same crop (e.g., corn at the end of the rotation followed by corn to start the rotation).

The figure was modified to clarify this point.

[Figure]

L167: Please provide more details about how the soil was processed, for example sieving and drying. How were the SOC stocks calculated?

Text added to clarify this point in L.175:

"The samples were oven-dried at 40 °C, ground and sieved to less than 2 mm before being analyzed to determine carbon content... "

And L.180:

"The bulk density measured in 2021 for the 0-20 cm layer of each plot of the analyzed treatments was used for the calculation of SOC stocks (R: 1.28 ± 0.02 Mg m-3; CC: 1.38 ± 0.03 Mg m-3)."

L173: Please provide more details about how the soil was prepared. Were they dried before incubations? Were the soils "pre-incubated" to avoid the CO2 flush from disturbance?

Yes, the samples were dry before incubation and were pre-incubated to avoid effects of the CO2 flush from disturbance.

Text added in L. 185:

"We incubated oven-dried soil (40°C) in hermetic glass bottles at 25 °C and with a moisture content equal to 60% of the soil field capacity to promote heterotrophic respiration. A pre-incubation was carried out to avoid effects of the $CO_2$ flush from disturbance."

L189: Were these C values adjusted to the KrCr2O7 method values as was done with the LECO C values?

We made a change regarding this point, which was detailed in the second response to Reviewer 1. POM-MAOM data was corrected in the following way:

- The measured POM data was corrected by a coefficient of 0.81.

- The MAOM fraction was calculated as the difference between soil bulk C and POM fraction. This result was then corrected by 0.81.

These recalculated POM-MAOM ratios were used then to run the modeling again. This did not modify the results and conclusions of the work.

L201: I recommend replacing the term "global average" with term "overall average," as "global average" could be misinterpreted as a worldwide average rather than am average within the dataset.

Changed according to the reviewer's suggestion.

L203 (Table 2): The horizontal line between soybean and pastures can be removed.

Changed according to the reviewer's suggestion.

L203 (Table 2): While it is necessary to use literature values to estimate total C inputs, the authors may wish to consider and discuss some of the shortcomings. In particular, using the shoot:root ratio presumably will give an estimate for total root inputs, but this study only focuses on the top 20 cm. In addition, for perennial species, the root system does not turn over every year,

and therefore the root:shoot ratio will overestimate root inputs. A value of 0.5 yr-1 for relative root turnover would be more realistic (e.g., Gill & Jackson 2000), except in the third pasture year when pasture is replaced by grain crops, and thus the entire root system turns over.

The reviewer is correct in highlighting that the estimation of C inputs is one of the weak points of this work. However, the depth of the profile evaluated in this study (20 cm) was considered when estimating the C inputs. The first author made an error by not explicitly stating this in the methodology before, but the relationship CPOM(0-20) / CPOM(0-80cm) = 0.723 (data not shown) was used to estimate the proportion of belowground inputs corresponding to the top 20 cm of the profile. This is in column S (Below_input_0-20) of the 'C_inputs' sheet in the DATA.xlsx file of the supplementary material. A description of this procedure is added to the methodology.

Based on the reviewer's suggestion to use a more realistic estimate of root turnover for pastures, all parameter adjustments for R system and all simulations were rerun considering a recalculated C input value. Following the reviewer's suggestion, a root turnover of 0.5233 was used, which is the average of this variable for all temperate zone grasslands reported in Gill and Jackson (2000). This value was used for all years of the pastures, except the last year in which all biomass was considered as input. Due to the low relative weight of the change made, the C input in the R System changed from 5.2 ± 1.2 to 4.89 ± 1.2 Mg ha$^{-1}$.

This change led to modifications in all numerical results for the R System, as the model was re-parameterized considering the new C input value. However, these changes did not affect the overall results and conclusions.

Text added in L.210:

"The proportion of belowground C inputs corresponding to the 0-20 cm layer was calculated by considering a coefficient of 0.72, obtained from the ratio between C POM (0-20 cm) and C POM (0-80 cm) for the site (data not shown) and assuming that the vertical distribution of roots correlates positively with the vertical distribution of C POM. In the case of pastures, a root turnover of 0.52 y$^{-1}$ (Gill and Jackson, 2000) was considered, except for the last year of the pasture phase for which a root turnover equal to 1 y$^{-1}$ was considered."

L215 (Equation 1): Has this model been used previously (e.g., Spohn et al. 2023)? If so, please provide a citation for reference.

Sentence added in L.238:

"This structure of a two-pool compartmental model with a connection in series has been used in works such as Spohn et al. (2023) and Stoner et al. (2021)".

L216, elsewhere: The term "amount" is ambiguous. Presumably, this refers to the mass of C. Please use explicit terms such as "mass."

Changed throughout the text according to the reviewer's suggestion.

L216, elsewhere: I believe that "k1 and k2" should be called "rate constants" rather than "rates." In contrast, "rates" are the product of the pool sizes and rate constants (e.g., k2*Cs is the rate of loss from the Cs pool). Being correct and consistent with the language throughout the manuscript will help the reader follow through these concepts.

We have modified the terms as suggested throughout the manuscript. Coefficient rates are assumed to be constant in the model of first order kinetics used in this work, although we should keep in mind that decomposition and retention coefficient rates can in fact vary (i.e. Soil C saturation theory, e.g.: Hassink and Witmore (1997), Stewart et al. (2007), Kemanian and Stocke (2010)).

L221: Please list and define the variables used in Figure 3 within the figure legend (e.g., I = system C inputs, Cl = POM C storage, etc.).

Figure 3 changed according to the reviewer's suggestions.

[Figure]

L229: Is Fa mass fraction or atom fraction? Please be explicit for reproducibility purposes.

Is the atom fraction. Text added in line 250.

L234: Should this be listed as Eq. 3?

Modification added.

L234: Please explain what the "-25" after "sample" and "-19" after "OX" mean, or else omit them from the equation.

Explanation added in L.253.

"By convention, the OX-I standard is normalized to a $\delta^{13}C$ value of -19‰, and the measured sample to a value of -25‰ to correct for mass-dependent isotopic fractionation effects (Trumbore et al., 2016)."

L246-247: Please give more details about how the priors were set. For example, was the prior normal or uniform?

Text added in L.272:

"For the MCMC procedure, a uniform distribution with a wide range of variation was used as a prior for each parameter, such that the outcome would be strongly dependent on the data."

L276: I think a title such as "Measured C dynamics" would be a better section title than "Measured data."

Title changed according to reviewer suggestion.

L284: The phrase "Regarding the isotopic information" can be removed for brevity.

Modification done.

L296: The units for oxidation rate seem to be missing the basis. For example, is this mg C hr-1 g-1 soil or mg C hr-1 microcosm-1?

The reviewer is correct. The variable 'Oxidation rate (mg C h-1)' was changed to 'Oxidation rate - microcosm (mg C h-1 microcosm-1)', and a new column was created in Table 3, 'Oxidation rate (mg g-1 C h-1)', which corresponds to the oxidation rate value standardized per unit mass of incubated C.

**Table 2.** Carbon stocks in fractions (POM, MAOM), radiocarbon in bulk soil and incubation efflux, and oxidation rates in incubations for each depth and agricultural system.

| System | Depth (cm) | Bulk soil $\Delta^{14}C$ (‰) | | Incubation efflux $\Delta^{14}C$ (‰) | Oxidation rate - microcosm (mg C h$^{-1}$ microcosm$^{-1}$) | Oxidation rate (mg g$^{-1}$ C h$^{-1}$) | POM stock (Mg ha$^{-1}$) | MAOM stock (Mg ha$^{-1}$) |
|---|---|---|---|---|---|---|---|---|
| | | 2008 | 2021 | 2021 | 2021 | 2021 | 2021 | 2021 |

| | | | | | | | |
|---|---|---|---|---|---|---|---|
| R | 0 - 10 | -34.17 [a] (3.57) | -47.27 [a] (9.54) | 9.13 [a] (2.98) | 0.0553 [a] (0.0029) | 0.080[a] (0.0052) | 5.55 [a] (0.48) | 22.40 [a] (0.23) |
| R | 10 - 20 | -46.9 [a] (9.9) | -45.97 [a] (2.97) | 48.3 [a] (5.56) | 0.0463 [a] (0.0033) | 0.089[a] (0.0034) | 1.16 [a] (0.12) | 20.90 [a] (0.79) |
| R | 0 - 20 | -40.0 [a] (6.47) | -46.69 [a] (6.65) | 27.1 [a] (4.82) | 0.051 [a] (0.0006) | 0.084[a] (0.00098) | 6.71 [a] (0.58) | 43.3 [a] (1.01) |
| CC | 0 - 10 | -91.5 [a] (17.60) | -60.4 [a] (3.71) | -6.5 [b] (3.15) | 0.0407 [b] (0.041) | 0.083[a] (0.0012) | 3.76 [b] (0.32) | 17.8 [b] (0.94) |
| CC | 10 - 20 | -107.7 [b] (16.86) | -86.47 [b] (2.85) | 25.63 [b] (3.77) | 0.0287 [b] (0.029) | 0.074[b] (0.0017) | 0.64 [b] (0.12) | 16.9 [b] (0.91) |
| CC | 0 - 20 | -99.37 [b] (17.24) | -72.11 [b] (3.32) | 6.87 [b] (3.09) | 0.035 [b] (0.0021) | 0.079[b] (0.00037) | 4.40 [b] (0.29) | 34.7 [b] (1.84) |

Note: Different letters within the same variable and depth indicate significant differences by paired t-test ($p < 0.05$) between systems. The value in parentheses indicates the standard error of the mean (n = 3). 'Oxidation rate – microcosm' corresponds to the total incubated material (25 g of soil); 'Oxidation rate' is standardized per unit mass of incubated C.

L297-298: This seems to be a repeat of the methods and does not need to be stated here.

That text was removed, and the result of the C input rates was moved to line 323.

L299: A reference to Table 5 may be appropriate here.

Modification done.

L310: "Once the parameterization procedure reached convergence (25000 iterations)" can be changed to "In the parameterized model" for brevity.

Modification done.

L318-319: The term "low" requires context, and the "% of outflow rate" needs to be defined. For example, "the transfer of POM to MAOM was small relative to the total POM outflow (i.e., MAOM transfers plus decomposition) ..." It may also be helpful to indicate that this implies that most of the POM was being respired rather than becoming MOAM.

Changes in line 349 to correct this point:

"The mean values for α were very low in all cases. This means that the transfer from Cfast to Cslow pool was small relative to the total Cfast outflow (i.e., transfers to Cslow plus decomposition). The mean values for α were 1.58 ±

0.00027%, 1.69 ± 1.77%, and 2.46 ± 2.15% of the C outflow rate from the fast pool in the R system, Period 1 of the CC system, and Period 2 of the CC system, respectively, which implies that most of the Cfast pool (~ POM) was being respired rather than flowing to the Cslow pool (~ MAOM)."

L320/309: Table 4 is redundant with Figure 4. Specifically, Table 4 is a summary of the information displayed in Figure 4. I recommend moving Table 4 to the supplemental, as the important information (means and distributions) can be seen in Figure 4.

Modification done. Table 4 is now in the Appendix (Table A2).

L320 (Figure 4): It is difficult to compare the k2 and a parameters between R and CC treatments because the x-axes are not consistent. The inset diagram helps, but I still find it difficult to grasp when the x-axes are different. I suggest expanding the x-axes for the R treatment to match those of the CC treatment.

The scale of the x-axis was modified to ensure it matches between treatments.

[Figure]

L330/335 (Figures 5 and 6): According to the methods, bulk soil samples were collected to a depth of 20 cm until 1996 and then were collected to a depth of 15 cm afterward. However, the SOC stock data presented here are all shown to a depth of 20 cm. Presumably, the measured 0-15 cm SOC stocks were extrapolated (e.g., multiplied by 20/15) to obtain an estimate for 0-20 cm stock. For transparency, please provide these details in the methods, and indicate this in the figure legend.

Yes. What the reviewer suggests is correct. Text added at L.170 of Materials and Methods to clarify this point:

"Taking this into account, we extrapolated the information on SOC stock measured from 1996 onwards up to a depth of 20 cm by multiplying by a coefficient equal to 20/15."

L330/335: These figures can be combined into one four-panel figure, as most readers will be interested in directly comparing the R and CC treatments.

Figures modified according to the reviewer's suggestion.

We also modified the figure in response to reviewer 2 comments to clarify that what we modeled was the dynamics of two conceptual compartments ('fast' and 'slow' pool) whose sizes were estimated based on the measurements of POM and MAOM in the year 2021.

[Figure]

L330/335: The colors used for bulk soil, MAOM, and POM should be consistent among panels.

Figures modified according to the reviewer's suggestion (shown in previous point).

L351 (Table 5): These terms should be defined in the table legend, particularly "stabilization flux" and "stabilization efficiency." It may also be more intuitive to change the term from "release" to "output." E.g., C input rate, total C output

rate, POM C output rate, MAOM C output rate. The "stabilization flux" may be better termed "POM-to-MOAM transfer."

All these changes have been incorporated into Table 5 (Table A3 in the new document).

**Table A3.** Main C flux rates derived from the compartmental model fitted to each agricultural system (for steady state condition for R system and for 2021 in CC system)

|  | R | CC |
|---|---|---|
| C input rate (Mg ha$^{-1}$ y$^{-1}$) | 4.94 (1.9) | 2.87 (1.1) |
| Total C output rate (Mg ha$^{-1}$ y$^{-1}$) | 4.94 (2.17E-05) | 3.00 (2.4E-02) |
| $C_{fast}$ output rate (Mg ha$^{-1}$ y$^{-1}$) | 4.86 (1.3E-03) | 2.81 (6.3E-02) |
| $C_{slow}$ output rate (Mg ha$^{-1}$ y$^{-1}$) | 0.078 (1.3E-03) | 0.20 (6.2E-02) |
| $C_{fast}$-to-$C_{slow}$ transfer rate (Mg ha$^{-1}$ y$^{-1}$) | 0.078 (1.4E-03) | 0.052 (6.2E-02) |
| Stabilization efficiency (%) | 1.57 (0.025) | 1.85 (2.2) |

Note: The value in parentheses indicates the standard deviation of the mean. Stabilization efficiency: $C_{fast}$-to-$C_{slow}$ transfer rate / C input rate

L351 (Table 5): Perhaps this information would be better displayed graphically by making a figure similar to Figure 3, except there would be a panel for R and a panel for CC. The arrow widths could be proportional to flows and the box sizes could be proportional to stocks (Cl and Cs). This isn't strictly necessary but would probably help users better understand the system differences.

Following the reviewer's suggestion, Table 5 was moved to the Appendix (Table A3), and Figure 5 was created to graphically represent the state variables and flux rates of the adjusted model for each agricultural treatment.

[Figure]

**Figure 5.** $C_{fast}$ and $C_{slow}$ stocks and main C flux rates derived from the compartmental model fitted to each agricultural system (steady state condition for R system and for 2021 in CC system)

L354: It would be ideal to define "age" and "transit time" here, even if they are defined in the methods, to remind the reader. I also recommend explaining what is meant by "system age."

The paragraph mentioned by the reviewer was modified to clarify these points. Changes were introduced in lines 386 and 392 to redefine the concepts of 'mean system age' and 'transit time'.

L355, elsewhere: This seems like too many significant figures, given that these results are based on a large data/model fusion, with significant uncertainty. For example, 545.65 ± 17.61 can probably be rounded to 546 ± 18.

Changes incorporated according to the reviewer's suggestion.

L368: In panel 7d, can the authors explain why the mean transit times are disparate even though the density distributions look nearly identical? Is there something happening with the density distributions that we cannot see in the graph?

Yes. Although in the graph presented in Fig. 7 (Fig. 6 in the new document) the transit time distributions appear similar in both systems, there is an "aging" effect of the output C flux of the R System compared to the CC when both are

evaluated at equilibrium. The R System has a $C_{slow}$ pool (MAOM) much larger than the CC at equilibrium and with a much older mean age (Fig. 6c), which determines that the transit time distributions of both systems are different at their right end (determined by the contribution of the $C_{slow}$ pool) but very similar at their left end (determined by the contribution of the $C_{fast}$ pool – POM -). To illustrate this point, the 99th percentile of the transit time distribution of the CC System at equilibrium corresponds to a value of 97 years, while in the case of the R System it is 282 years. However, these differences are only noticeable at the right end of the distributions, making it impossible to represent in Figure 7d.

Text added at the end of the discussion section (L. 546):

"Additionally, it is important to highlight the difference in mean transit time between treatments (R: 11.5 years; CC: 6.2 years), even though the left end of their distribution, shown in figure 6d, is very similar. The R System has a slow-cycling pool much larger than the CC at equilibrium and with a much older mean age (Fig. 6c), which determines that the transit time distributions of both systems are different at their right end (determined by the contribution of the slow pool) but very similar at their left end (determined by the contribution of the fast pool). This difference explains why the mean transit times are disparate even though the distributions look nearly identical."

L385-387: This is a bit concerning, given that the model does not account for erosion. Moreover, even the comparison of SOC stocks through time assumes that there is no erosion. Is there any estimate for erosion rates at this site that could be presented here? I would think that erosion rates would be low, given that the slope is 3%, but it would be helpful to have a ballpark estimate.

Indeed, this is one of the main weaknesses of the work, particularly in the CC System. However, Baethgen et al. (2021), based on simulations with the CENTURY model coupled with RUSLE, suggest that of the total C outputs, around 20% have been due to erosion in the CC System and 13% in the R System. It was not possible to incorporate this process in the relatively simple modeling approach used in this work, but based on this previous information, it is clear that the vast majority of the observed differences are due to differences in the process of SOC oxidation between treatments. If we were to make the simplistic reasoning of deducting this proportion of outputs due to erosion from the calculated k2 for both treatments post hoc, the decomposition rate of the Cslow pool (discounting erosion) would be between 3.4 and 4.9 times higher in CC than in R. Regardless, it is clear that the explicit incorporation of the erosion process in models representing the 14C fate is an important issue that should be addressed in future works with more availability of isotopic data.

Text was added at L419-425 to clarify this point.

"Regarding this, it is important to highlight that tillage interventions were progressively reduced throughout the experiment starting from 1983, and soil management completely changed to no-till in 2009 in both systems. Therefore, erosive processes should have been more relevant in Period 1 (1964-1990) but significantly less in Period 2 (1991-2021). A previous study on the same experiment (Baethgen et al., 2021) suggests that the vast majority of SOC losses are due to heterotrophic respiration in both treatments (around 87% for the R system and 80% for the CC system). However, future studies should consider this process, explicitly separating SOC losses and evaluating their impact on radiocarbon dynamics and its use in model parameterization."

L408-467: I think this discussion section would be clearer if the authors used language that clearly differentiated pool dynamics (i.e., $C_l$ and $C_s$) versus flux dynamics (i.e., I, a, $k_1$, $k_2$). This would help the reader relate back to the underlying model (Fig. 3). For example, the R and CC system had a similar a and $k_1$ flux constants, but the R system had approximately twice as much C input, which lead to the POM pool being 17% greater and the total flux of C from POM to MOAM to be greater. It would also be helpful to use different terminology when referring to rate constants (e.g., $k_1$) versus rates or fluxes (e.g., $(1-a)*k_1*C_l$).

The terminology used in the discussion was modified to clearly differentiate rate constants from rates.

L415-421: This seems somewhat contradictory. The first part states that litter quality is important, and the second part states that maybe litter quality did not matter. From this study, where litter quantity and quality were confounded, I think the conclusions are that 1) POM was greater in system with higher/better litter 2) modeled (not measured) POM flux dynamics (proportional transfer and decomposition) did not vary by system. However, since litter quality and quantity were confounded, and litter quality was not explicitly modeled, I do not think the model provides insight into whether litter quality impacted the pools and fluxes.

What was attempted to be explained from line 420 (old document) is the following:

- The quality of the input is a determining factor in the dynamics of SOC in general, and particularly of the more labile SOC (Córdova et al., 2018).
- We did not find significant differences in the dynamics of the fast pool (POM) between treatments ($k_1$).
- This could be connected to the fact that the grain cropping phase of the R system has exactly the same crops as CC, and in the pasture phase,

although there is a mix of grasses and legumes, residues with low C/N may only represent a relatively small proportion of the inputs in time when considering the whole period.

The reviewer is correct in noting that this reasoning is essentially speculative, so the following text is added to provide context to the previous assertions (L.469):

"However, the quality of the C input was not explicitly considered in this model, and therefore, it is confounded with the effect of quantity. Hence, caution should be taken in assessing the causes and implications of the absence of differences between the $k_1$ outflow rate constants of both treatments."

L430: Should "periods" be "treatments?" If not, please clarify this sentence.

It was necessary to adjust the CC model in two stages to account for the different slopes of SOC stock decline. The "Period 1" spans from 1964 to 1990, and "Period 2" from 1991 to 2021. This is explained at the beginning of section 3.2 Model results. To make it clearer, the meaning of 'Period 1' and 'Period 2' was clarified in parentheses where the reviewer indicated.

L439-440, L452-453: This is an important point that may warrant additional attention, for example in the abstract. That is, even though POM contributed relatively little to the differences in SOC stocks, these model results suggest that having a relatively large POM pool is a prerequisite for forming and/or maintaining MOAM.

In response to the reviewer's suggestion, this point was emphasized in the conclusions and in the abstract.

L442-445: While I agree with this explanation in principle, it seems like this would apply to both the R and CC systems. Are there other mechanisms that could explain why this was only seen in the CC system?

The explanation starting at L.438 (old document) applies to the CC system since the low C input rate and the smaller size of its fast pool (POM) can determine that the slow pool (MAOM) has a greater role as a nutrient provider for microbial biomass, increasing its oxidation.

The process outlined starting from L.442 (old document) also has greater relevance in the CC system given that the frequency of soybean cultivation in the CC treatment is twice that in the R System. In the CC System, the crop rotation repeats every three years, while the same crop rotation in the R System is in three out of every six years (the other three are pastures). This

higher frequency of alternation between soybean and crops with high C:N ratio could explain part of the differences observed between treatments.

Text added in L.500 to clarify this point:

"The frequency of soybean crop is twice as high in the CC system as in the R system (Fig. 1), so this process could explain part of the differences observed between treatments."

L460-461 vs. 466-467: These statements about litter quality again seem somewhat contradictory. The first states that the stabilization was not a function of litter quality, but the second states that stoichiometry (i.e., quality) could have an effect. Overall, these contrasting ideas should be rectified throughout the discussion.

The text was modified at the end of the 'MAOM accrual hypothesis' section to make it clearer that, according to the literature, there may be an effect of input quality on the dynamics of C stabilization, but this was not explicitly evaluated in this work.

(L.516) "As suggested by previous studies, different qualities in the C input between treatments could also be relevant in determining the stabilization processes previously discussed (Ma et al., 2018; Zhu et al., 2020; Córdova et al., 2018). However, the model used in this study does not explicitly represent the influence of input quality, which becomes confounded with the effects of quantities. Therefore, in this case, we can assert that the higher C stabilization process in integrated crop-pasture systems in R seems to be essentially explained by the quantity of C inputs, and there may be an effect of the quality that should be studied more deeply."

L469: "Global Properties (Transit Time and Age)" can be changed to "C transit time and age."

Modification done.

L483: The world "globally" can be removed to avoid confusion with "worldwide."

Modification done.

L493-494: This sentence needs clarification. Specifically, it is unclear what is meant by "rejuvenation" effect. If anything, losing old MAOM seems like the opposite of rejuvenation.

In the CC system, the MAOM pool ($C_{slow}$) has a much higher outflow rate constant compared to the R System. This means that new inputs to this pool are increasingly less diluted by an old MAOM pool (because this pool is progressively smaller). This process determines that the MAOM pool progressively becomes younger in the CC System, as the mass of recently entered C has a greater relative weight (because the entire mass of that pool is smaller).

Text added (L.562): "However, it is clear that the CC system induces a "renewal" effect by significantly losing very old C from the $C_{slow}$ pool which progressively becomes more diluted with , the inputs to this compartment through transfers from the $C_{fast}$ pool."

L504-505: As the quality of litter was confounded with quantity, and quality was not explicitly modeled, the statement that litter quality was not important should not be a central conclusion.

The reviewer is correct. That conclusion was removed, and the following text was added (L584):

"Although no differences were observed in the model parameters that determine the C flow to the $C_{slow}$ pool (~ MAOM) per unit of C entering the system, it was not possible to establish a mechanistic connection with the quality of the C input, given that this factor is not explicitly modeled."

Additional text was added to the conclusions (L.587):

"Regarding the implications of these results on the management of agricultural systems, it is important to maintain a high rate of well-distributed C inputs over time to feed the microbial community, preventing the destabilization of old C. Therefore, the central importance of reducing fallow periods and increasing system productivity to raise C inputs becomes relevant, not only to enhance the stabilization processes of new C but mainly as a mechanism to prevent the loss of old C."

3. REVIEWER 2

The manuscript "High capacity of integrated crop-pasture systems to preserve old stable carbon evaluated in a 60-year-old experiment" is an interesting study of how different crop management practices can influence the carbon cycling in soils. The long-term nature of the study is unique and provides nice insight into these processes.

We are very grateful for the valuable comments raised by the reviewer. Below, we present the answers to each of the points made. In many cases, these responses led to modifications in the original manuscript, so we outline the lines (corresponding to a new version of the manuscript) where these modifications were introduced.

General comments:

The term "stable" is pretty loaded, I suggest the authors use a different term. Stability is being interchanged with decreased decomposition and with MAOM, which I don't think is always true. From the model, the observed differences in 14C and C stock of bulk soil are likely due to higher decomposition rates in the CC system, which doesn't necessarily indicate higher stability at the RR sites. I don't think the authors have shown that the RR system C is more stable, just that the RR system increases C stocks by reducing losses. If the RR site were to be tilled in a similar manner to the CC system, would you expect the C stock value to persist, or would it decrease similar to the CC sites? If the latter, I don't think it's fair to say the C is stable, just that the land management decreases losses. The authors sort of get at this in lines 464-465.

Following this, I think it would be more appropriate to refer to the modeled pools as something like "fast cycling" and "slower cycling", rather than calling them "POM/labile" and "MAOM/stable". We know that some MAOM doesn't persist for very long, so it is misleading to interchange the terms. Modifying word choice does not impact the conclusions of the model, which are quite interesting and provide nice perspective on SOC cycling rates in the two pools. (As an aside, it would be interesting to see the 14C of the POM and MAOM that the authors physically separated and how this matched up to the modeled pools, though I realize this may be outside the scope of this project).

The reviewer is correct regarding this point. It is not possible to draw conclusions about the inherent stability of the MAOM in both agricultural systems, as the dynamics of this pool in CC and R emerge as a consequence of each of the agronomic managements performed. Therefore, the term "stability" would not be correct in this context. Based on the reviewer's

suggestion, the terms "fast cycling" and "slow cycling pool" were adopted to refer to each of the compartments throughout the manuscript.

Regarding the 14C data in fractions, they are indeed interesting data but beyond the scope of this manuscript. It is expected that this information will be available later, and will contribute then to deepen the understanding of these systems in future work.

Interpretation of the incubation data: I disagree with the statement that the incubation CO2 from the CC system is "more modern" than that of the R system (Lines 289-293). In the 0-10 cm, the CC system incubation CO2 is -6.5 permil and the R is 9.13 permil, making the R system CO2 more modern. All of the other results are 14C modern (positive numbers falling on the bomb curve). You can't distinguish which side of the curve they are on and therefore can't claim one is more modern than the other.

The reviewer is correct; this statement is clearly incorrect. Given the results of the modeling work and reconsidering the 14C data, the slightly lower isotopic signature in the $CO_2$ output from the CC system incubations is probably due to a greater contribution from the MAOM pool in the output flow, which leverages the average delta14C towards lower values.

The text was modified at L. 321 to eliminate the comparison of the "ages" of C that was mistakenly made when preliminarily analyzing the 14C results in the incubations.

"$CO_2$ radiocarbon measured in soil incubation experiments from 2021 samples showed significant differences between systems at all depths (Table 3), with values of 6.87± 3.09 ‰ in the CC system (0-20 cm) and 27.1 ± 4.82 in the R system. In both cases, these were much more modern (closer to the atmospheric signature of the year of measurement) than the bulk soil."

Title: I suggest rewording this a bit following the above comments on wording

Based on the reviewer's suggestion, the manuscript title was changed to:

"High capacity of integrated crop-pasture systems to preserve old soil carbon evaluated in a 60-year-old experiment", deleting the word "stable".

Technical/minor edits:

-Lines 31: Using the phrase "on the one hand" makes it sounds like the two perspectives are in opposition, but I don't think these are.

Text changed according to reviewer suggestion.

-Line 117: should be "acidic"

Text changed according to reviewer suggestion.

-Line 146: clarify if fertilizer application was using in the crop-pasture rotation system

Yes, the R system is also fertilized. Text was added at L.161 to clarify this point:

"In both systems, fertilization (N and P) of crops and pastures is carried out according to recommendations based on soil and plant analysis."

-Line 172: how many is "a large number"?

This internal laboratory validation process was conducted with over 2000 soil samples.

-Line 185: what was the actual temperature? (> 500 C)

The actual temperature was 550°C. The text was modified to address this suggestion from the reviewer.

-Line 229: The "a" in "Fa" should be subscript

Modification made.

-Lines 499 and 501: "Ancient" is a bit of a stretch for this age of carbon

This word was replaced by "old" where the reviewer mentions.

-Lines 503-505: This sentence is speculative and not actually tested in the study.

To correct the speculative statement made about the influence of C inputs on the dynamics of Cslow (~ MAOM) formation, the text was changed where the reviewer suggests (at line L. 566):

"This process may occur to a greater extent in the integrated system due to higher carbon inputs, which may result in a larger stock of labile C and

consequently greater microbial activity. Although no differences were observed in the model parameters (k1 and α) that determine the C flow to the slow-cycling pool per unit of C entering the system, it was not possible to establish a mechanistic connection with the quality of the C input, given that this factor was not explicitly modeled."

---

## Author Response (AR2)

Author's response

We appreciate once again the opportunity to continue with the review process of our work. Below is a detailed description of the changes made to the article at this stage.

In Figure 6, panels b and c, it may be helpful to include the word "age" on the X axis.

Figure 6 was modified according to the suggestion.

L523-525: It would be helpful to provide a more explicit interpretation regarding the management of the pasture. Is it typical for a pasture to be mowed without any biomass removal in an agricultural rotation? If not, then it should ideally be pointed out that this type of management is atypical, and that additinal work should be done under more typical management practices (e.g., biomass is removed for hay).

Text was added in L.525 referring to this point:

"This management was implemented in this experiment to facilitate operational aspects by avoiding the inclusion of direct grazing by animals. Since typical management of these agricultural systems producing beef and dairy would include either the presence of grazing animals in pasture fields, or the removal of biomass produced by hay harvest, future work should consider these effects on the dynamics of SOC, in an approach of compartmental model adjustment assisted by isotopic information."

L536-537: It is not clear what the is the comparison in this statement: "...the mean transit time reached lower values..." What are the values lower than?

Text was modified from L.543 to answer to this point:

However, the mean transit time in CC system reached lower values than in R system, though these differences were not statistically significant (CC system: 6.2 years; R system: 11.5 years) (Fig. 6d). This is explained by the different dynamics of the slow pool, which, because of higher output rates in CC system, tends to become younger as the new C entering this compartment is less diluted in a large stock of old C.

L550: Please clarify this: "...of which more than half (13.5%) ..." 13.5% is less than half.

Text was slightly modified from L.556 to answer to this point:

Focusing on the age structure of C in the R system, we found that 21.4% of its C (12.10 Mg ha$^{-1}$) was incorporated during the period of the experiment (59 years), of which more than half (7.58 Mg ha$^{-1}$) consists of fast-cycling C (~ POM), a pool that in its entirety is younger than the age of the experiment. The remainder of the total renewed C corresponds to slow-cycling C (~ MAOM), and only 8.8% of the C from this pool entered the system after the year 1963; therefore, most of this pool was already present in the soil before the experiment's installation.